# Green synthesis of graphite from $CO_2$ without graphitization process of amorphous carbon

Chu Liang [1,2,3,5], Yun Chen [1,5], Min Wu [1], Kai Wang [1], Wenkui Zhang [1✉], Yongping Gan [1], Hui Huang [1], Jian Chen [4], Yang Xia [1], Jun Zhang [1], Shiyou Zheng [2✉] & Hongge Pan [3✉]

Environmentally benign synthesis of graphite at low temperatures is a great challenge in the absence of transition metal catalysts. Herein, we report a green and efficient approach of synthesizing graphite from carbon dioxide at ultralow temperatures in the absence of transition metal catalysts. Carbon dioxide is converted into graphite submicroflakes in the seconds timescale via reacting with lithium aluminum hydride as the mixture of carbon dioxide and lithium aluminum hydride is heated to as low as 126 °C. Gas pressure-dependent kinetic barriers for synthesizing graphite is demonstrated to be the major reason for our synthesis of graphite without the graphitization process of amorphous carbon. When serving as lithium storage materials, graphite submicroflakes exhibit excellent rate capability and cycling performance with a reversible capacity of ~320 mAh g⁻¹ after 1500 cycles at 1.0 A g⁻¹. This study provides an avenue to synthesize graphite from greenhouse gases at low temperatures.

[1] College of Materials Science and Engineering, Zhejiang University of Technology, Hangzhou 310014, China. [2] School of Materials Science and Engineering, University of Shanghai for Science and Technology, Shanghai 200093, China. [3] School of Materials Science and Engineering & State Key Lab of Silicon Materials, Zhejiang University, Hangzhou 310027, China. [4] Institute of Science and Technology for New Energy, Xi'an Technological University, Xi'an 710021, China. [5]These authors contributed equally: Chu Liang, Yun Chen. ✉email: msechem@zjut.edu.cn; syzheng@usst.edu.cn; hgpan@zju.edu.cn

Elemental carbon has been extensively applied in the interdisciplinary fields spanning catalysis, metallurgy, environmental remediation, energy storage and conversion, automotive industry, and drug delivery because of its tunable physicochemical properties[1–6]. Although carbon is the fourth most abundant element in nature by mass, more than 99% of carbon appears in the form of compounds such as metal carbonates, organics, carbides, and carbon dioxide/monoxide[7–12]. The controllable synthesis of elemental carbon from carbon containing compounds has become the major strategy of achieving carbon materials with various physicochemical properties[7–10,13,14]. Carbon atoms bond together in different ways to form carbon allotropes with different physicochemical properties. Graphite, the most thermodynamically stable allotrope under standard conditions, has attracted special attention owing to its excellent physicochemical properties, including electrochemical lithium storage, electric and thermal conduction, superlubricity, and chemical and thermo stability[7,15–17].

Graphite can be separated from natural graphite mine or synthesized from carbon containing compounds. The separation of natural graphite requires multistep procedures including graphite mining and large-scale beneficiation and purification, which is the complex and inefficient method of production[18]. In the purification procedure, a large amount of hydrofluoric acid is expended to remove the mineral impurity, which devastates our natural environment. Further intensive purification is needed to produce the battery-grade graphite for lithium-ion batteries. The total material loss is as high as ~70% for producing natural graphite. The separation of natural graphite is a time-consuming and environmentally unfriendly process. Moreover, the microstructure and morphology of natural graphite are largely dependent on its natural deposits.

Synthetic graphite, as a type of crystalline carbon with tunable microstructure and morphology, of which the synthesis procedures generally contain two sequential processes: carbonization of carbon precursors and graphitization of amorphous carbon[19–22]. During the carbonization of carbon precursors such as biomass and organic materials, considerable quantities of greenhouse gas ($CO_2$) and hazardous gases (e.g., CO, $SO_2$, and $NO_x$), which are one of the main causes of global warming and environmental pollution, are emitted into the atmosphere. After carbonization, the carbon precursors are converted into graphitizable or non-graphitizable carbon. Direct graphitization of graphitizable carbon at high temperature (~3000 °C) and catalytic graphitization of non-graphitizable carbon at a temperature of ~1000 °C are the two primary routes of transforming amorphous carbon into graphite[22,23]. Moreover, the transition metal catalysts are found to be difficult to separate from synthetic graphite[24]. The green and efficient synthesis of graphite with controllable microstructure and morphology remains a considerable challenge.

In this work, we explore a green, ultralow-temperature, and efficient route to synthesize graphite with controllable microstructure and morphology from $CO_2$ without the graphitization process of amorphous carbon. The $CO_2$ is converted into graphite submicroflakes within 3 s as the mixture of $CO_2$ and lithium aluminum hydride (LiAlH$_4$) is heated to 126 °C, which is the lowest temperature for synthesizing graphite up to now. As-synthesized graphite submicroflakes, serving as anode materials for lithium storage, are demonstrated to show excellent rate capability and cycling performance with a reversible capacity of ~320 mAh g$^{-1}$ at 1.0 A g$^{-1}$ after 1500 cycles.

## Results

**Synthesis and characterization.** Figure 1a shows a schematic illustration of the synthesis of graphite. In the absence of transition metal catalysts, $CO_2$ is directly converted into graphite without the graphitization of amorphous carbon at high temperatures. The variation of sample temperatures and gas pressures with time in the synthesis process was recorded in Fig. 1b. In the initial stage, the sample temperature and gas pressure increased linearly with time at a constant heating rate of 2 °C min$^{-1}$. When the $CO_2$–LiAlH$_4$ sample is heated to 126 °C, the sample temperature jumps to 876 °C in 3 s, implying exothermic nature of the reactions between $CO_2$ and LiAlH$_4$. A steep increase in gas pressure with time is simultaneously observed on account of dramatic change in temperature. The value for gas pressure changes is 36 bar in the wide temperature range of 126–876 °C, which is much less than 57 bar of linear pressure changes in the narrow temperature range of 35–126 °C, indicating that considerable amounts of $CO_2$ are consumed to react with LiAlH$_4$. After the removal of impurities in the solid products of exothermic reactions, the as-obtained black powder is confirmed to be graphite (Fig. 1c, Supplementary Figs. 1, 2, and Supplementary Table 1). It is a remarkable fact that the synthesis temperature of our graphite is the lowest one reported to date and the reaction time is the shortest one as well[20–22].

As shown in Fig. 1c, the strong and sharp X-ray diffraction (XRD) peak at 26.36° can be assigned to the (002) plane of graphite. The lattice spacing of d (002) is calculated to be 3.38 Å, in accordance with bulk graphite[22,25], suggesting a high degree of graphitization of graphite. This conclusion can also be drawn from the Raman spectrum (Fig. 1d). The intensity ratio of strong G band around 1582 cm$^{-1}$ to weak D band around 1350 cm$^{-1}$ is as high as ~3.9. Besides, a strong 2D band around 2703 cm$^{-1}$, corresponding to highly ordered graphitic carbon[26], is observed in the Raman spectrum of graphite. The as-synthesized graphite can be further demonstrated to be high degree of graphitization by XPS (X-ray photoelectron spectroscopy) spectra (Fig. 1e, f). A very small amount of O was detected in the as-synthesized graphite, in which O is chemical bonding with C. For the C element, the intensity of sp$^2$-C peak is much greater than that of sp$^3$-C peak as shown in the high-resolution XPS spectrum of C 1s. Figure 2 presents the SEM, TEM, HRTEM, and SAED images of as-synthesized graphite. It exhibits the flake shaped morphology with a thickness of 160–350 nm (Fig. 2a, b). The submicroflake shaped morphology can be observed in the TEM and HRTEM images (Fig. 2c, d). The d-spacing of graphite submicroflakes determined by the HRTEM image is about 0.333 nm (Fig. 2d), in accordance with the XRD result above and the values of graphite previously reported[22]. The HRTEM image and SAED pattern (Fig. 2e) further confirm the formation of graphitic phase. The specific surface area of graphite submicroflakes is 14.5 m$^2$ g$^{-1}$ (Supplementary Fig. 3), calculated by the Brunauer-Emmett-Teller model.

In contrast to the high-temperature graphitization and catalytic graphitization, the as-synthesized graphite is produced by reacting $CO_2$ with LiAlH$_4$ at low temperatures less than 876 °C for only several seconds (Fig. 1b). Our synthesis of graphite consumes $CO_2$, whereas the above two methods of creating synthetic graphite produce $CO_2$ and hazard gases during the carbonization of precursors. $CO_2$ has been reported to synthesize graphitic carbon sheets by molten salt electrolysis[27] or thermal reaction of $CO_2$ with CaC$_2$[28]. However, the graphitization degree of graphitic carbons is much less than our as-synthesized graphite since the intensity ratio of G band to D band of 3.9 is far greater than that of 1.7 of graphitic carbon synthesized by molten salt electrolysis at 850 °C and that of 1.3 of graphitic carbon synthesized by reacting $CO_2$ with CaC$_2$ at 700–800 °C. During the synthesis of above graphitic carbons, the evolution of CO is accompanied. In this work, the synthesis of graphite is a green, and time-saving process. Furthermore, our as-synthesized graphite is easy to separate from impurities or byproducts as

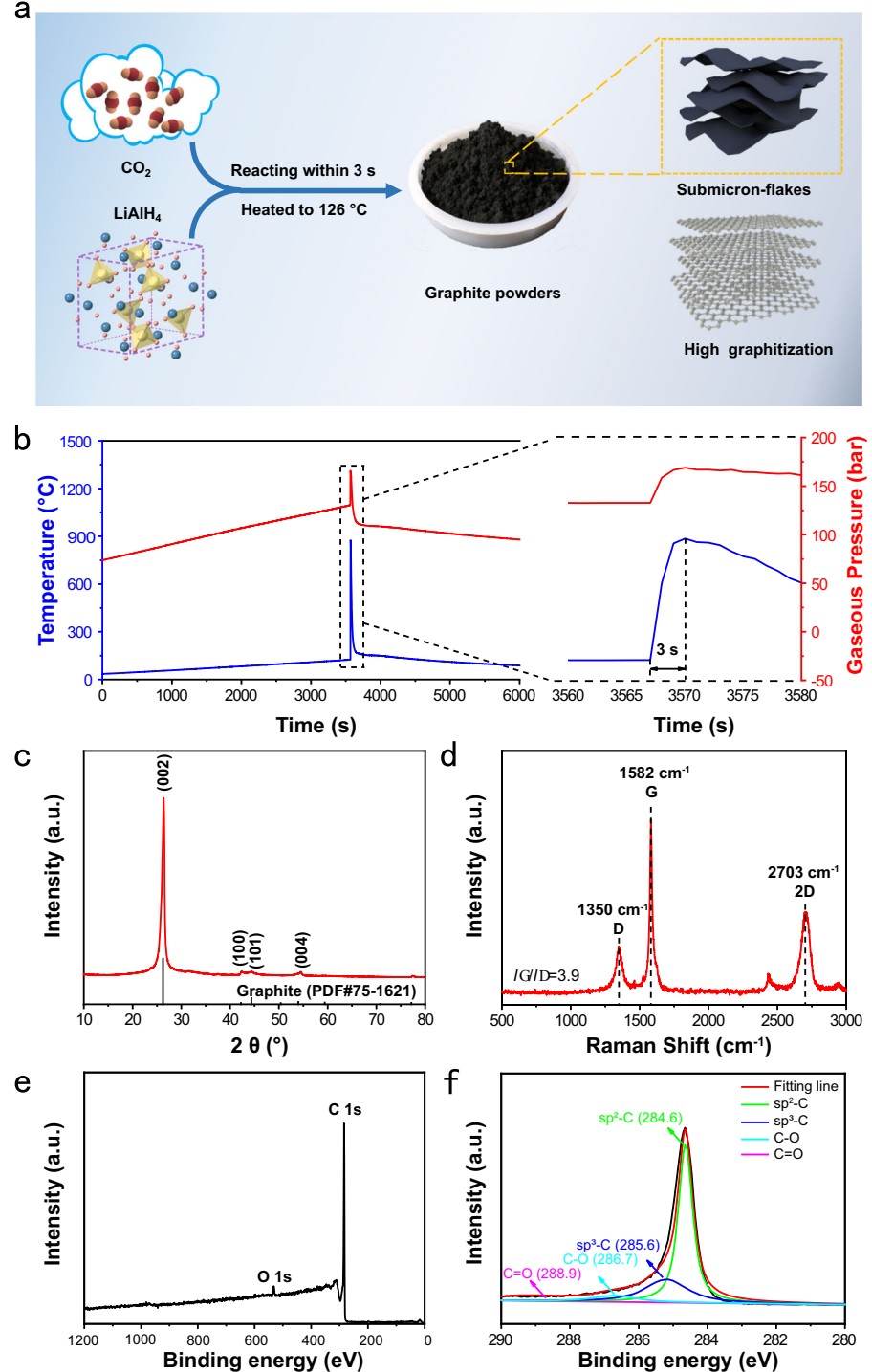

**Fig. 1 The synthesis and characterization of graphite derived from CO₂. a** Schematic illustration of the synthesis of graphite submicroflakes. **b** Time dependence of temperature and gas pressure in the reactor during the reaction process. **c** XRD pattern of the solid products after reaction. **d** Raman spectrum of the solid products after the removal of impurities. **e** XPS survey spectrum of graphite submicroflakes. **f** High-resolution XPS spectrum of C 1 s.

indicated by energy dispersive spectroscopy (EDS) analysis and thermogravimetric (TG) measurement (Supplementary Figs. 1 and 2). The content of metal ions in as-synthesized graphite was characterized by inductively coupled plasma-mass spectrometry (ICP-MS) (Supplementary Table 2). The purity of as-synthesized graphite is 99.988 wt% (metals basis), very close to 99.996 wt% of commercial graphite (Supplementary Table 2).

**Conversion reaction of graphite from CO₂.** The XRD pattern of the solid products of $CO_2$ reacting with $LiAlH_4$ is illustrated in Fig. 3a. The strong characteristic XRD peaks of $LiAlO_2$, $Li_2CO_3$, and Al are seen in the XRD pattern, signifying the chemical interaction between $CO_2$ and $LiAlH_4$ in the exothermic process (Fig. 1b). According to the Rietveld refinement result, the weight ratio of $LiAlO_2$ to $Li_2CO_3$ is calculated to be 77:23, equal to a

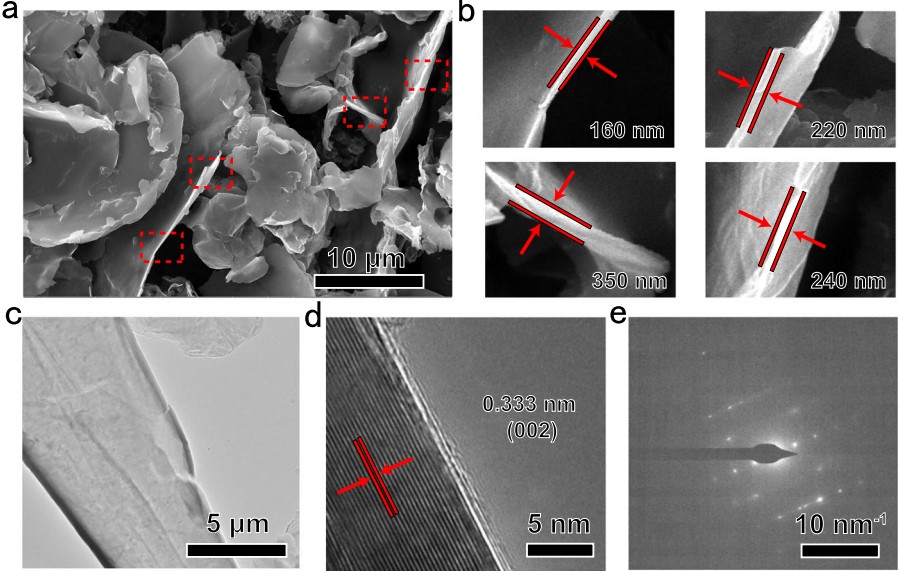

**Fig. 2 Morphology and microstructure of graphite submicroflakes. a, b** SEM images. **c** TEM image. **d** HRTEM image. **e** SEAD pattern.

molar ratio of 2.97:1. The weight ratio of Al to $LiAlO_2$ or $Li_2CO_3$ is inaccurate on the basis of the Rietveld refinement method as serious particle aggregation of Al is observed in the above solid products (Supplementary Fig. 4 and Supplementary Table 3). In addition, the XRD peaks of graphite can be identified in Fig. 3a, but its intensity is lower than that of other crystalline phases in solid products. This may be resulted from the fact that graphite is insensitive to X-ray compared with Al, $LiAlO_2$, and $Li_2CO_3$.

H, $H_2$, $CO_2$, CO, C, and O signals were detected in the gaseous products of $CO_2$ reacting with $LiAlH_4$ (Fig. 3b). The observation of H and $H_2$ signals implies the generation of hydrogen in the synthesis process of graphite. H signal originates from the decomposition of $H_2$ during the mass spectrum (MS) measurement. The excessive $CO_2$ is the main reason for the detection of $CO_2$, CO, C, and O signals in the MS of gaseous products. The MS signals will be the same as Fig. 3b if CO was produced in the synthesis process of graphite. CO can be identified in the gas mixture of CO and $CO_2$ by Fourier transform infrared (FTIR) spectra. As shown in FTIR spectra (Fig. 3c), the characteristic absorption of $CO_2$ is observed in the wavenumber range of 2250–2400 $cm^{-1}$, whereas the absorption of CO is not seen in the wavenumber range of 2000–2250 $cm^{-1}$. The FTIR and MS results indicate that CO is not produced in the above synthesis process and hydrogen is the only new developed gas product.

Based on the phase identification of solid products and composition analysis of gaseous products, the chemical interaction between $CO_2$ and $LiAlH_4$ can be described by the following equation:

$$10LiAlH_4 + 9CO_2 \rightarrow 7C + 6LiAlO_2 + 2Li_2CO_3 + 4Al + 20H_2,$$
(1)

where the theoretical molar ratio of $LiAlO_2$ to $Li_2CO_3$ is very close to the experimental value of 2.97:1 determined by Rietveld refinement of XRD patterns. The theoretical mass ratio of solid products to reactants is 1.94:1, which is more than the experimental value of 1.84:1. This result is attributed to the evaporation of Al at the temperature of 660 °C, above its melting point (Fig. 1b). The decreasing Al content is calculated to be 3.3 wt% owing to the evaporation. To further confirm the chemical reaction (1), the solid products were subjected to TG analysis in the air. The weight change in solid products resulted from the reaction of air with carbon and Al, and the decomposition of

$Li_2CO_3$ is clearly seen in the TG curves (Supplementary Fig. 5). The total weight loss is 10.3 wt%, consistent with theoretical value of 10.8 wt%, in which 3.3 wt% Al loss and Eq. (1) are used in the calculation of theoretical value. The heat released from the chemical reaction (1) is calculated to be 733.6 kJ $mol^{-1}$ C based on the standard formation enthalpies of reactants and products[29], in accordance with the exothermic nature as seen in Fig. 1b. These results further demonstrate that the chemical interaction between $CO_2$ and $LiAlH_4$ can be expressed by Eq. (1). According to chemical reaction (1), the theoretical yield of graphite based on $LiAlH_4$ is calculated to be 22.1 wt%, which corresponds closely to the experimental value of 20.7 wt% determined by the TG measurement of solid products of $CO_2$ reacting with $LiAlH_4$ (Supplementary Fig. 5). The yield of graphite is as high as 93.7%.

**Formation mechanism of graphite derived from $CO_2$.** Graphite is the most thermodynamically stable allotropic form under standard condition. However, carbon with low graphitization degree (amorphous carbon) is easy to produce in the carbonization of traditional carbon precursors[7–12], due to the very high kinetic barrier for the formation of graphite. The graphitization of amorphous carbon at high temperatures is an indispensable procedure in order to synthesize graphite[21,22,24,30]. Our synthesis method is different from the above two-stage synthetic procedure. The conversion reaction from $CO_2$ to graphite can occur in the absence of transition metal catalysts when the $CO_2$–$LiAlH_4$ system is heated to as low as 126 °C, indicating low kinetic barrier of the conversion reaction (1) for synthesizing graphite. The favorable thermodynamics and kinetics are responsible for the graphite being directly formed from $CO_2$ at the temperature range of 126–876 °C within several seconds (Fig. 1b).

The solid products of $CO_2$ reacting with $LiAH_4$ (Fig. 1b) were heated at 880 °C for 3 h under argon. Few differences in graphitization degree is observed in the as-synthesized carbon with and without heat treatment at 880 °C (Supplementary Fig. 6), signifying that the amorphous carbon was not converted into graphite at 880 °C in the presence of $LiAlO_2$, $Li_2CO_3$, and Al. For understanding the graphitization of amorphous carbon, first-principles calculation was employed to calculate the kinetic barriers (Supplementary Fig. 7). The activation energy is as high as 1.66 eV, supporting that amorphous carbon was not converted into graphite at 880 °C and higher temperatures are needed for

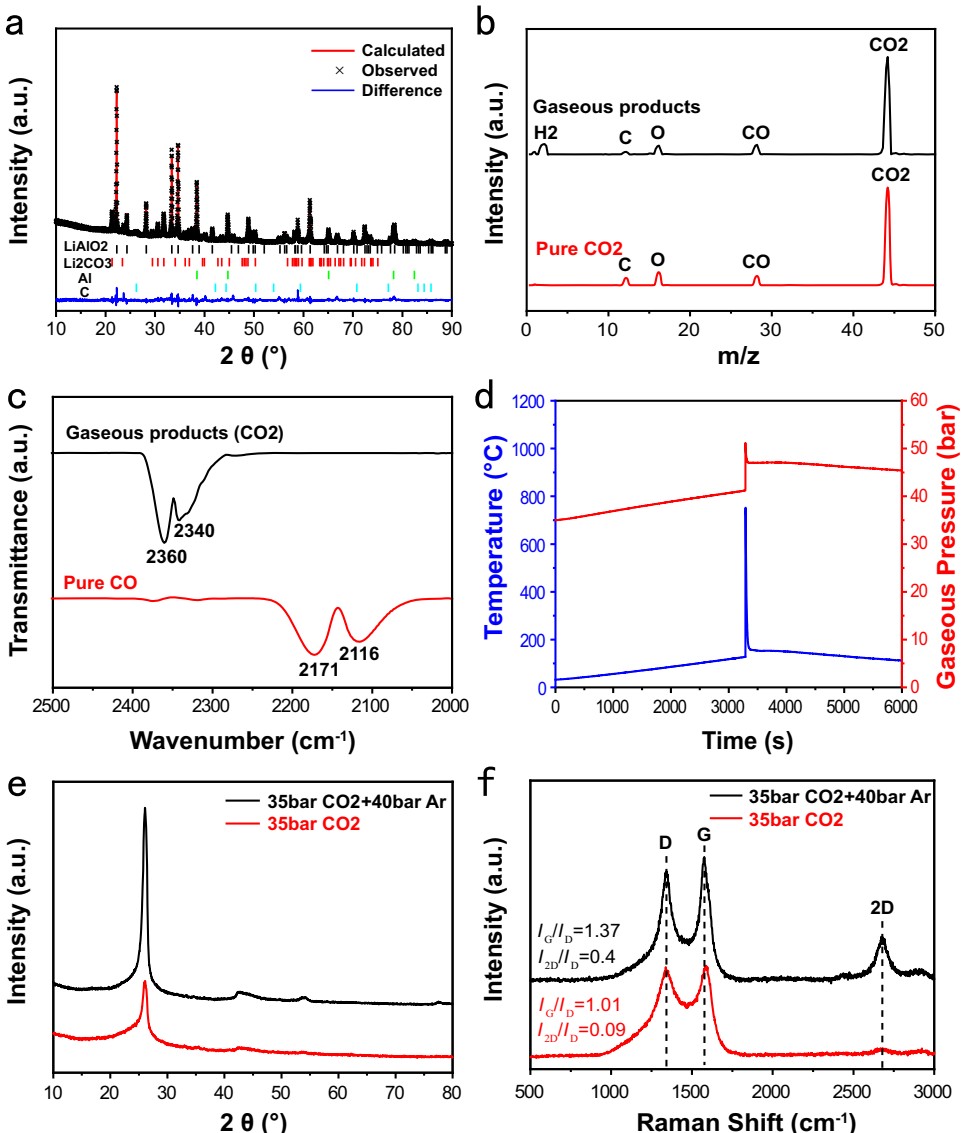

**Fig. 3 Characterization of the products of $CO_2$ reacting with LiAlH$_4$. a** Rietveld refinement of the XRD pattern of the solid products of $CO_2$ reacting with LiAlH$_4$. **b** The mass spectrum of the gaseous products and pure $CO_2$. **c** FTIR spectrum of gaseous products and pure CO. **d** Time dependence of temperature and gas pressure in the reactor during 35 bar $CO_2$ reacting with LiAlH$_4$. **e** XRD patterns and **f** Raman spectra of carbon synthesized by reacting 35 bar $CO_2$ with LiAlH$_4$ under the gaseous back pressures of 35 and 75 bar.

the graphitization of amorphous carbon. In other words, our graphite is directly synthesized by reacting $CO_2$ with LiAlH$_4$. The reaction kinetics for synthesizing graphite derived from $CO_2$ is found to strongly depend on the $CO_2$ pressure. As shown in Fig. 3d, the sudden changes in temperatures and gas pressure are also observed in the heating process when the initial gas pressure of $CO_2$ was decreased to 35 bar, indicating an exothermic reaction of $CO_2$ with LiAlH$_4$. The maximum temperature is about 752 °C, lower than the one of 876 °C as shown in Fig. 1b, suggesting that reaction kinetics is associated with $CO_2$ pressure. For this as-synthesized carbon, the (002) peak of graphite at 26.36° and a broad and weak peak in the range of ~17–26° are clearly seen in the XRD pattern (Fig. 3e). Two partially overlapped G and D bands and one 2D band are observed in the Raman spectrum of the carbon synthesized under 35 bar $CO_2$ (Fig. 3f). The intensity ratio of G band to D band is 1.01, which is much less than the 3.9 of graphite submicroflakes. The 2D band centered at 2703 cm$^{-1}$ supports highly ordered graphitic carbon. The above XRD and Raman results imply that the carbon synthesized under 35 bar

$CO_2$ is the mixture of graphite and amorphous carbon. Compared to graphite submicroflakes (Fig. 1), it can be concluded that the content of graphite in as-synthesized carbon decreases with the $CO_2$ pressure applied in the synthesis process resulted from the reaction kinetics of synthesizing graphite related to $CO_2$ pressure.

To further examine the effect of $CO_2$ concentration and gas pressure on the reaction kinetics for synthesizing graphite, inert argon was first introduced into the 35 bar $CO_2$–LiAlH$_4$ reactor until the gas pressure of 75 bar was reached. During heating, $CO_2$ reacted with LiAlH$_4$ as sudden changes in temperatures and gas pressure were detected (Supplementary Fig. 8). The as-synthesized carbon exhibits stronger XRD peaks of graphite and weaker XRD peaks of amorphous carbon than the carbon synthesized under 35 bar $CO_2$ (Fig. 3e), further supporting that more graphite is produced in the carbon synthesized under higher gaseous back pressure. This can also be concluded from the Raman spectrum (Fig. 3f). The intensity ratios of G band to D band and 2D band to D band are calculated to be 1.37 and 0.40, respectively, greater than 1.01 and 0.09 of the carbon synthesized

under 35 bar $CO_2$. When the $CO_2$ pressure increases to the value as shown in Fig. 1b, the intensity ratios of G band to D band and 2D band to D band increase to as high as 3.9 and 1.8, respectively, for the graphite submicroflakes. It can be concluded that the kinetic barriers of the reaction for synthesizing graphite decrease with the gaseous back pressure. In addition, $CO_2$ concentration has little effect on the kinetic barriers of the reaction for synthesizing graphite (Supplementary Fig. 9) in contrast with gaseous back pressure.

Temperature is another key factor in synthesizing graphite from $CO_2$. Take the $LiAlH_4$ reacting with 35 bar $CO_2$ as an example, various reaction temperatures are achieved by adjusting the amounts of $LiAlH_4$ in the synthetic experiments. As the amounts of $LiAlH_4$ are increased from 0.3 g to 0.38 and 0.5 g, the maximum temperature increases from 471 °C to 752 and 960 °C, respectively (Supplementary Fig. 10). The 2D band is clearly seen in the Raman spectrum of the carbon synthesized at the maximum temperature of 960 °C (Supplementary Fig. 11), whereas it is not observed in the Raman spectra of the carbon synthesized at the maximum temperatures of 471 and 752 °C (Supplementary Fig. 11). Moreover, their intensity ratios of G band to D band are 0.88, 1.01, and 1.47, respectively, signifying graphitization degree of carbon increases with temperature. The same conclusion can be drawn from the XRD patterns (Supplementary Fig. 11). The above Raman and XRD results indicate that the graphitization degree of carbons strongly depends on their temperatures. However, the kinetic barrier of a reaction is independent of temperature. To overcome the kinetic barriers, heating is a good way for reactants system to absorb energy. According to our first-principles molecular dynamics (FPMD) calculations, the total energy of $CO_2$ significantly increases with the temperature (Supplementary Fig. 12). This is the reason for the temperature-dependent synthetic reaction of graphite.

In order to analyze the reason for a small amount of amorphous carbon formed in the synthesis of graphite, carbon was synthesized by reacting $LiAlH_4$ with 2 bar $CO_2$ (Supplementary Fig. 13). The starting and maximum reaction temperature are 142 and 165 °C, respectively. The starting temperature is higher than the one in the high gaseous back pressures, but the maximum temperature is much lower than that in the high gaseous back pressures (Figs. 1b, 3d and Supplementary Fig. 8). The characteristic peaks of graphite are not observed in the XRD pattern of as-synthesized carbon (Supplementary Fig. 14). The G band almost overlaps the D band, and the 2D band is not seen in the Raman spectrum of the carbon synthesized under 2 bar $CO_2$ (Supplementary Fig. 15). The XRD and Raman results indicate that amorphous carbon is synthesized but graphite is not synthesized under 2 bar $CO_2$, resulted from the low $CO_2$ pressure-induced high kinetic barriers of the reaction for synthesizing graphite. The thermodynamic and kinetic competition of the reaction for synthesizing graphite and amorphous carbon as illustrated in Supplementary Fig. 16 leads to the synthesis of graphite from $CO_2$, and it strongly depended on $CO_2$ pressure and temperature. The kinetic barriers of the reaction for synthesizing graphite drop with rising of gaseous pressure. On the other hand, the kinetic barriers of the reaction for producing amorphous carbon is insensitive to gaseous pressure because the starting reaction temperature only decreased from 142 °C under 2.2 bar $CO_2$ to 126 °C under 132 bar $CO_2$ (Fig. 1b and Supplementary Fig. 13). Even under the high pressure of ~160 bar, the kinetic barriers of the reaction for synthesizing graphite are slightly higher than those for producing amorphous carbon while it is thermodynamically favorable to produce graphite.

First-principles calculation based on density function theory was performed to study the $CO_2$ pressure-dependent kinetics and thermodynamics of the reaction for synthesizing graphite. The energy difference between graphite and amorphous carbon is calculated to be −0.57 eV, supporting the favorable formation of graphite from the thermodynamic theory. A slight increase in total energy with a large increase in pressure is observed for both graphite and amorphous carbon (Supplementary Fig. 17). The total energy of the reactants of $CO_2$ and $LiAlH_4$ remarkably increases with the pressure, particularly for $CO_2$ (Supplementary Fig. 17). The density of $CO_2$ is dependent on its pressure due to the isometric process applied to synthesize graphite. For the $CO_2$–$LiAlH_4$ system under low $CO_2$ pressure, an appreciable reduction in the total energy of $CO_2$–$LiAlH_4$ system can be obtained by a small amount of $CO_2$ adsorbing on the $LiAlH_4$ surface (Supplementary Fig. 18). In order to achieve high pressure in the synthesis of graphite, gaseous $CO_2$ was compressed into a liquid, where the increased energy of $CO_2$ resulting from work done on gas approximately equals to the liquefaction heat of $CO_2$ of 0.16 eV[31]. The liquid $CO_2$ is converted into gaseous state during heating, in which the energy of $CO_2$ absorbed is equivalent to the vaporization heat of 0.16 eV[31]. The great increase in energy of 0.32 eV is the additional energy for the graphite synthesized under low $CO_2$ pressure. Overall, the $CO_2$ pressure-induced energy changes of reactants and products are responsible for the $CO_2$ pressure-dependent kinetics and thermodynamics of the reaction for synthesizing graphite.

**Electrochemical lithium storage performance.** Figure 4a shows the cyclic voltammogram curve of graphite submicroflakes. A typical lithium storage behavior of graphite is seen in the CV curve of graphite submicroflakes. The reduction peaked at ~1.15 V, which is also observed in the CV curves of commercial graphite (Supplementary Fig. 19), emerging at the initial cycle but disappearing in the following cycles. The 2nd cycle of CV curves overlap the 3rd cycle, implying a stable solid electrolyte interphase (SEI) layer formed on graphite electrodes in the initial cycle[30]. The pair of peaks at ~0.1 and ~0.25 V vs. Li/Li+ is the reversible redox peaks of graphite. The graphite submicroflakes deliver a reversible capacity of 343 mAh g−1 at 0.1 A g−1 with an initial Coulombic efficiency of 77.5% (Fig. 4b), lower than 90% of the commercial graphite (Supplementary Fig. 20). The 7.9 m2 g−1 increase in specific surface area (Supplementary Fig. 3) leads to increased irreversible capacity of graphite submicroflakes in the initial cycle (Fig. 4b and Supplementary Fig. 20). Both graphite submicroflakes and commercial graphite exhibit stable reversible capacities around 320 mAh g−1 from 1st to 100th cycles at 0.1 A g−1 (Supplementary Fig. 21). After 100 cycles, the capacity retention of graphite submicroflakes is 99%, higher than 95.4% of commercial graphite.

The rate capability of graphite submicroflakes is shown in Fig. 4c, d. The reversible capacity of graphite submicroflakes is much higher than that of commercial graphite when the cells discharge/charge at the current density above 0.1 A g−1. At the current densities of 0.1, 0.5, 1.0, 2.0, and 4.0 A g−1, the graphite submicroflakes deliver the reversible capacities of 349, 296, 247, 172, and 82 mAh g−1, respectively, corresponding to the capacity retention of 94%, 80%, 66%, 46%, and 22% relative to 372 mAh g−1 of the theoretical capacity of graphite. The reduction in reversible capacities of graphite at higher current densities is due to the shortened lithiation/delithiation voltage plateau induced by discharge/charge kinetic performance (Fig. 4d and Supplementary Fig. 22). Compared to graphitized mesocarbon microbeads (MCMBs), one of the main anode materials in lithium-ion batteries, the graphite submicroflakes still exhibit higher reversible capacities and superior rate capability (Fig. 4c). After the rate capability measurement as shown in Fig. 4c, the cells were used to further test the cycling performance at 1.0 A g−1 (Fig. 4e). The graphite submicroflakes

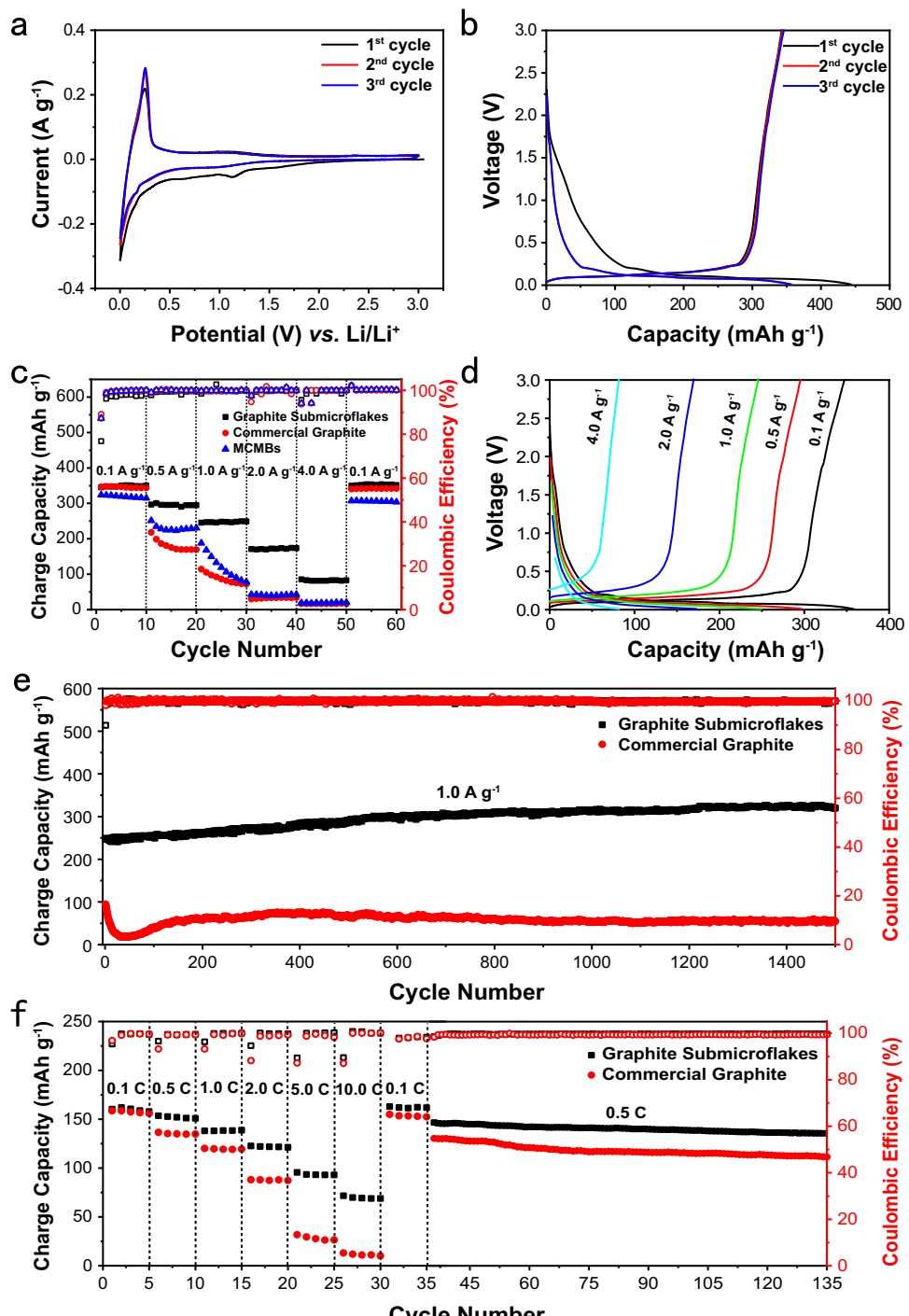

**Fig. 4 Electrochemical lithium storage performance of graphite. a** Cyclic voltammogram curves of graphite submicroflakes cycled between 0.01 and 3.0 V. **b** Discharge/charge curves of graphite submicroflakes at a current density of 0.1 A g$^{-1}$. **c** Rate capability of graphite submicroflakes and commercial graphite from 0.1 to 4.0 A g$^{-1}$. **d** Discharge/charge curves of graphite submicroflakes from 0.1 to 4.0 A g$^{-1}$. **e** Long-term cycle performance of graphite submicroflakes and commercial graphite at 1.0 A g$^{-1}$. **f** Rate capability of full cells at 0.1C–10C (1C = 170 mA g$^{-1}$).

deliver an initial reversible capacity of ~250 mAh g$^{-1}$ at 1.0 A g$^{-1}$. The cycling capacity slowly increases to ~320 mAh g$^{-1}$ at the first 1200 cycles and then maintains the maximum capacity even up to 1500 cycles. The reversible capacity of graphite submicroflakes at the 1500$^{th}$ cycle is 5.9 times higher than 54 mAh g$^{-1}$ of commercial graphite.

The graphite submicroflakes anode was coupled with LiFePO$_4$ cathodes to evaluate the rate capability and cycling performance of graphite in full cells (Fig. 4f). The full cell delivers a reversible

capacity of ~160 mAh g$^{-1}$ at 0.1C (1C = 170 mA g$^{-1}$), almost equal to that of LiFePO$_4$/commercial graphite full cells. The difference in the rate capability increases with the discharge/ charge current densities for the full cells with the graphite submicroflakes anodes and commercial graphite anodes. At the current density of 10C, it delivers a reversible capacity of ~69 mAh g$^{-1}$ for the LiFePO$_4$/graphite submicroflakes full cells and ~10 mAh g$^{-1}$ for the LiFePO$_4$/commercial graphite full cells, corresponding to the capacity retention of 43% and 6%,

respectively. The excellent rate capability and stable cycling performance are achieved in the LiFePO₄/graphite submicroflakes full cells.

## Discussion

In summary, we have demonstrated a method for the green synthesis of graphite from $CO_2$ at low temperatures. The graphite submicroflakes are successfully synthesized by reacting $CO_2$ with $LiAlH_4$ at the temperature range of 126–876 °C within several seconds in the absence of transition metal catalysts. The kinetic barrier of reaction for synthesizing graphite is found to strongly depend on gaseous back pressure in the synthesis process. The graphite and amorphous carbon mixture in different weight ratios is synthesized via tuning $CO_2$ pressure. The gaseous-back-pressure-induced thermodynamic and kinetic competition of the reaction for synthesizing graphite and amorphous carbon are responsible for the direct formation of graphite and amorphous carbon in various weight ratios. When used as anode materials for lithium-ion batteries, as-synthesized graphite submicroflakes show excellent rate capability and cycling performance because of their unique microstructure and morphology. The graphite submicroflakes deliver a reversible capacity of ~320 mAh g⁻¹ after 1500 cycles at the current density of 1.0 A g⁻¹.

## Methods

**Materials and synthesis**. High-purity deionized (DI) water was prepared in our lab. The other chemical reagents were obtained from the commercial purchase and used as received. The graphite submicroflakes are synthesized by the chemical interaction between $CO_2$ and $LiAlH_4$ in a home-made stainless-steel reactor setup (Supplementary Fig. 23). The reactor setup is mainly composed of a sample cell with a sample temperature monitor and a gas reservoir with temperature and pressure monitors. The volume is 30 and 20 ml, respectively, for the sample cell and the gas reservoir. The sample cell is connected to the gas reservoir by gas line. The temperatures and gas pressures can be recorded by an online data collection system. For each synthetic experiment, 0.30, 0.38, or 0.50 g of $LiAlH_4$ (97%) purchased from Alfa Aesar were loaded into a sample cell in a MBRAUN glovebox ($O_2$<0.5 ppm, $H_2O$<0.5 ppm) filled with argon (99.995%, Outesen). Then, gaseous $CO_2$ (99.995%, Pujiang) and/or argon was introduced into the above reactor setup. Various preset gaseous pressures were applied in the experiments. To produce a high pressure above 100 bar in the heating process, liquid $CO_2$ was introduced into the above reactor via gas injection system (Supplementary Fig. 23) because gaseous $CO_2$ was converted into the liquid above its critical pressure of 73.8 bar. Valves 1 and 3 were opened in the $CO_2$ injection process and were closed as the $CO_2$ injection was finished. Valve 2 was kept close in the $CO_2$ injection process, whereas it was kept open to connect the reactor to gas reservoir in the subsequent heating and reaction process of $CO_2$ with $LiAlH_4$. The $CO_2$–$LiAlH_4$ mixture was heated at a rate of 2 °C min⁻¹ from room temperature to preset temperatures. The sample temperatures and gas pressures in the reactor setup were monitored by temperature sensors and pressure transducers, respectively. The solid reaction products between $CO_2$ and $LiAlH_4$ were collected to react with excess hydrochloric acid (37 wt%, Xilong) at 200 °C. Graphite was obtained by separating solid from liquid and washing with DI water. To remove the part of carbon with low degree of graphitization, the as-obtained graphite powders were mixed with KOH (ACS, Aladdin) at a mass ratio of 1:5 followed by heating at 850 °C for 3 h under nitrogen (99.995%, Outesen). Finally, the high-purity graphite submicroflakes were successfully synthesized after washing with DI water and ethanol (99.7%, Ante) and drying at 80 °C under vacuum.

**Characterization**. The products of $CO_2$ reacting with $LiAlH_4$ and graphite samples were characterized by XRD (X'Pert PRO), Raman spectroscopy (Renishaw Invia plus), field emission scanning electron microscopy (FESEM, NOVA NANOSEM 450) with EDS (Oxford X-Max 80 SDD), high-resolution transmission electron microscopy (HRTEM, FEI Tecnai G2 F30), ICP-MS (PerkinElmer Elan DRC-e), and XPS (Kratos Axis Ultra DLD) with monochromatized Al Kα excitation source. The XRD data were collected on an X'Pert Pro diffractometer with Cu Kα radiation at 40 kV and 40 mA in the 2θ range of 10–80°. Raman spectra were obtained at the excitation wavelength of 532 nm. Nitrogen adsorption and desorption measured on a Micromeritics ASAP 2020 were used to determine the specific surface area and pore size distribution of graphite submicroflakes. TG analysis (Q5000IR) was carried out from room temperature to 700 °C at a heating rate of 5 °C min⁻¹ in air. Gas composition was determined by gas chromatography-mass spectrometry and FTIR spectra (Thermo Nicolet 6700).

**Electrochemical measurements**. CR2032 coin-type cells were used to evaluate the electrochemical lithium storage performance of graphite, in which lithium foil (99.9%, China Energy Lithium) and lithium iron phosphate ($LiFePO_4$, battery grade, Dongyangguang) were selected as the counter electrodes for half cells and full cells, respectively. For the sake of contrast, commercialized graphite (99.8%, Alfa aesar) was used as the reference anode material. The anode electrode consisted of active material (85%), Super P (5%), and polyvinylidene fluoride (10%) and the cathode electrode consisted of active material (80%), Super P (10%), and polyvinylidene fluoride (10%) in N-methyl pyrrolidone to form slurry, which was coated on copper or aluminum foils evenly and followed by drying at 80 °C for 20 h in vacuum. The electrolyte was 1 M $LiPF_6$ dissolved in ethylene carbonate (EC), dimethyl carbonate (DMC) and diethyl carbonate (DEC) in a ratio of 1:1:1 by volume. Celgard membrane 2400 was used as the separator for electrochemical evaluation. Cyclic voltammetry (CV) tests were conducted on a CHI650B electrochemical work station. Galvanostatic discharge–charge and long-term cycle performance were measured using a Neware battery test system. The voltage ranges were 3.0–0.01 V for half cells and 3.8–2.0 V for full cells. Prelithiation was conducted for both graphite submicroflakes and commercial graphite. The electrochemical impedance spectra (EIS) were obtained on a ZAHNER from $4 \times 10^6$ Hz to $10^{-2}$ Hz. All electrochemical performance tests were conducted at the room temperature.

**Computational details**. All the total energy and molecular dynamics calculations were performed using the projector augmented wave (PAW) formalism of density functional theory (DFT) as implemented in the Vienna Ab-initio Simulation Package (VASP). The Perdew-Burke-Ernzerhof (PBE) generalized gradient approximation was employed for the exchange-correlation function. An energy cutoff of 400 eV for the plane-wave expansion of the wavefunctions was used for all the calculations. For geometry optimization, the atomic coordinates were relaxed until the Hellmann-Feynman forces were less than 0.01 eV/Å. The structural models of graphite (space group: P63/MMC), $LiAlH_4$ (space group: P2₁/C), $CO_2$ (space group: Pa-3), and amorphous carbon were employed to calculate the pressure-dependent energy differences (Supplementary Fig. 17). The structural model of amorphous carbon was quenched and optimized from the melted carbon at 6000 K with a supercell of 64 carbon atoms by FPMD calculation with the canonical ensemble (NVT). The Nose-Hoover thermostat was used to control the temperature. In order to calculate the adsorption energies of $CO_2$ molecules on the $LiAlH_4$ surface, a slab of $LiAlH_4$ with 6 layers of atoms was constructed with the bottom 3 layers of atoms being fixed without optimization to mimic the bulk structure. A vacuum layer with a thickness of 10 Å was constructed to avoid the interactions between the layers. The adsorption energies $E_{ad}$ were calculated from the following equation:

$$E_{ad} = E_{total} - E_{sub} - nE_{CO2}, \qquad (2)$$

where $E_{total}$ is the total energy of the systems, $E_{sub}$ is the energy of the substrate, $E_{CO2}$ is the energy of a free $CO_2$ molecule, and $n$ is the number of the adsorbed $CO_2$ molecules. On the other hand, FPMD calculations at 300, 400, 500, and 700 K were performed to study the temperature effect on the $CO_2$. At each temperature, MD simulation with a period of 10 ps and a time step of 1 fs was performed with the canonical ensemble NVT as well. A Nose-Hoover thermostat was used to control the temperature.

## Data availability

The data that support the findings of this study are available from the corresponding author upon request.

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

## Acknowledgements
This work was financially supported by National Natural Science Foundation of China (51677170, 52072342, 51671135, and 51971146) and Zhejiang Provincial Natural Science Foundation of China (LY19E010007). We acknowledge the support of the Program of Shanghai Subject Chief Scientist (17XD1403000), Shanghai Outstanding Academic Leaders Plan, and the Innovation Program of Shanghai Municipal Education Commission (2019-01-07-00-07-E00015). We also thank Dr. Chunsheng Wang at the Department of Chemical and Biomolecular Engineering of University of Maryland for giving the comments and suggestion on this paper.

## Author contributions
C.L., W.Z., S.Z., and H.P. conceived the ideas and revised the papers; C.L. designed the experiments; C.L. and Y.C. synthesized the materials and performed majority of the materials characterization; Y.C. and K.W. carried out the electrochemical test; Y.C., Y.G., and H.H. analyzed the microstructural data; Y.X. and J.Z. analyzed the electrochemical data; M.W. and J.C. designed the theoretical investigation on formation mechanism and performed the DFT calculations; and C.L. and Y.C. co-wrote the manuscript. All authors discussed and commented on the manuscript.

## Competing interests
The authors declare no competing interests.
