## [Peer Review File · Nature Communications]

REVIEWER COMMENTS

Reviewer #1 (Remarks to the Author):

This work reports a low temperature conversion of CO₂ to graphite by LiAlH₄ reduction, which is interesting not only for carbon capture but also for the lithium batteries therefore deserves a publication.

However, this referee has two concerns.

One is about the cost of this process. The yield of graphite based on LiAlH₄ has to be supplemented, which is an important parameter for a cost evaluation.

The other is about the reaction conditions. The discussion about the formation mechanism of graphite needs to be strengthened. The authors focused mainly on the role of high pressure, however, the influence of high temperature may be also very important. According to the authors' findings, all cases with graphite generated generally experienced a high temperature stage (752 or 865 C), while on the contrary, the reaction taking place at lower temperatures (between 142 and 165 C) did not produce any graphite at all, it is possible that the high temperature is the key for the graphite formation, and a recent literature reporting the conversion of CO₂ to graphite by thermo-reaction (Carbon 2017, 116, 246) indicated that the preferred graphite formation temperature is about 700 C.

Reviewer #2 (Remarks to the Author):

This paper reports graphite submicroflakes prepared from CO₂ via reacting with LiAlH₄ at 126 °C. It is looking very nice at a glance. However, there are some serious issues which are needed to be clarified properly. This work can be reconsidered after a mandatory revision.

1. Reduction of CO₂ will produce graphite. Theoretically it is correct. But the author is silent about yield. The author should mention about the amount of precursor liquid CO₂ and how much CO₂ (%) is converting to graphite (yield).

2. Even though the reaction temperatures are very low, gaseous pressure is too high. The advantage of mild synthesis condition is simple reactor design and low cost, but I could not find any reactor setup to sustain such a high pressure. The authors should clarify this issue related to high pressure.

3. The proposed mechanism in session 2.3 is not clear. For instance, the formation of ordered meso phase as intermediate was observed? More strong evidences such as computational and in-situ analyses are required to confirm the mechanism, which should be further clarified more detailed.

4. In abstract it is claimed that "CO₂ is converted into graphite submicroflakes in the seconds timescale via reacting with LiAlH₄ as the CO₂-LiAlH₄ mixture is heated to 126 °C"...

In conclusion it is claimed that "The graphite submicroflakes are successfully synthesized by reacting CO₂ with LiAlH₄ below 876 °C within"...

Which one is correct temperature 126 or 876 °C? It is recommended to clearly mention about the claims as the synthesis procedure is the main finding of this work.

5. There are some other reports (Studies in Surface Science and Catalysis, 1998, 114, 147-152, J. Mater. Chem. A, 2015, 3, 21211-21218) of converting CO₂ to graphite. The author is suggested to compare these synthesis methods and to highlight the novelty of this work.

6. It is mentioned in introduction "the transition metal catalysts are found to be difficult to separate from synthetic graphite" ... for other methods. However, it is not clear how pure graphite is obtained by the present technique, as the author also used LiAlH₄, KOH, hydrochloric acid etc.

7. How has the “carbon with low degree of graphitization” been removed with mixing with KOH?
8. The synthesis technique is quite confusing. In the main text, it mentioned that the graphite has been prepared in different pressure (35 bar and 75 bar) and temperatures. However, in the synthesis procedure it is not clearly reported about pressure and temperature.
9. It is recommended to further analyze the chemical structure of the as-synthesized material by XPS.

Responses to Reviewer 1#

Comment

This work reports a low temperature conversion of CO₂ to graphite by LiAlH₄ reduction, which is interesting not only for carbon capture but also for the lithium batteries therefore deserves a publication. However, this referee has two concerns.

Reply:

Thank you for your positive comment. We have revised our manuscript according to your comments and the detailed responses are given in the **Comments 1 and 2**.

Comment 1

One is about the cost of this process. The yield of graphite based on LiAlH₄ has to be supplemented, which is an important parameter for a cost evaluation.

Reply:

The yield of graphite based on LiAlH₄ has been added in the revised manuscript. According to TG measurement of the solid products of CO₂ reacting with LiAlH₄ heated under air, the yield of graphite based on LiAlH₄ is determined to be 20.7 wt.%. This experimental value is close to the theoretical value of 22.1 wt.% calculated from the chemical reaction (1) of CO₂ with LiAlH₄. The yield of graphite is as high as 93.7%.

For this synthesis process of graphite, the cost may be higher than the current industrial production of natural graphite and synthetic graphite. However, our synthesis of graphite is a green and energy-saving process, compared with the industrial production of natural graphite and synthetic graphite. As discussed in the introduction, the separation of natural graphite from graphite mine is a time-consuming, 70 wt.% of material-loss and environmentally unfriendly process. As for the synthetic graphite, on one hand, considerable quantities of greenhouse gas (CO₂) and hazardous gases (e.g. CO, SO₂ and NO_x) are emitted into the atmosphere in the carbonization of graphite precursors. On the other hand, high temperatures are

needed to perform the graphitization of amorphous carbon. The cost of the industrial production of graphite will be obviously increased if the cost of environmental remediation is involved. In this work, the greenhouse gas (CO_2) is used as a raw material to synthesize graphite and hydrogen is the only gaseous state product. Moreover, we report for the first time that the graphite is directly produced from CO_2 without the graphitization process of amorphous carbon. More work is needed to decrease the cost of this green and energy-saving synthesis of graphite.

The following content was added:

Page 11, Para 1, the end,

... further demonstrate that the chemical interaction between CO_2 and LiAlH_4 can be expressed by Equation (1). **According to chemical reaction (1), the theoretical yield of graphite based on LiAlH_4 is calculated to be 22.1 wt.%, which corresponds closely to the experimental value of 20.7 wt.% determined by the TG measurement of solid products of CO_2 reacting with LiAlH_4 (Supplementary Fig. 5). The yield of graphite is as high as 93.7%.**

Comment 2

The other is about the reaction conditions. The discussion about the formation mechanism of graphite needs to be strengthened. The authors focused mainly on the role of high pressure, however, the influence of high temperature may be also very important. According to the authors' findings, all cases with graphite generated generally experienced a high temperature stage (752 or 865 °C), while on the contrary, the reaction taking place at lower temperatures (between 142 and 165 °C) did not produce any graphite at all, it is possible that the high temperature is the key for the graphite formation, and a recent literature reporting the conversion of CO_2 to graphite by thermo-reaction (Carbon 2017, 116, 246) indicated that the preferred graphite formation temperature is about 700 °C.

Reply:

Thanks for your suggestion. We have discussed the effect of temperature and gaseous pressure on the synthesis of graphite in our revised manuscript. The

experiments on the effect of temperature and gaseous pressure on the synthesis of graphite have been supplemented. Moreover, the DFT calculation was employed to study the formation mechanism of graphite in the revised manuscript.

For a reaction, a lowest energy, generally defined as activation energy, is required to overcome the kinetic barriers with a certain value. In this work, the chemical reaction between CO₂ and LiAlH₄ has been demonstrated to be described by the following equation.

The role of high pressure is systematically discussed in the original version. It has been concluded that the kinetic barriers of the reaction for synthesizing graphite decrease with the gas pressure. Temperature is another key factor in synthesizing graphite from CO₂. The influence of temperature on the degree of graphitization of carbon was systematically investigated in the revised version. Take the LiAlH₄ reacting with 35 bar CO₂ as an example, various reaction temperatures are achieved by adjusting the amounts of LiAlH₄ in the synthetic experiments. As the amounts of LiAlH₄ are increased from 0.3 g to 0.38 and 0.5 g, the maximum temperature increases from 471 °C to 752 and 960 °C, respectively (Supplementary Fig. 10). The 2D band is clearly seen in the Raman spectrum of the carbon synthesized at the maximum temperature of 960 °C (Supplementary Fig. 11) whereas it is not observed in the Raman spectra of the carbon synthesized at the maximum temperatures of 471 and 752 °C (Supplementary Fig. 11). Moreover, their intensity ratios of G band to D band are 0.88, 1.01 and 1.47, respectively, signifying graphitization degree of carbon increases with temperature. The same conclusion can be drawn from the XRD patterns (Supplementary Fig. 11). The above Raman and XRD results indicate that the graphitization degree of carbons strongly depends on their temperatures. However, the kinetic barrier of a reaction is independent of temperature. To overcome the kinetic barriers, heating is a good way for reactants system to absorb energy. According to our first-principles molecular dynamics calculations, the total energy of CO₂ significantly increases with the temperature (Supplementary Fig. 12). This is the

reason for the temperature-dependent synthetic reaction of graphite.

As for the graphitic carbon synthesized by reacting CO_2 with CaC_2 at 700–800 °C (Carbon, 2017, 116:246), the intensity ratio of G band to D band is 1.3, which is much less than 3.9 of the graphite submicroflakes synthesized in this work. This Raman result implies relatively low graphitization degree of graphitic carbon synthesized by reacting CO_2 with CaC_2 at 700–800 °C. Moreover, CO_2 can react with 0.2 g LiAlH_4 to convert into graphite within 3 seconds under 145 bar maximum gaseous pressure, where the maximum temperature is as low as 595 °C. As shown in Raman spectrum, the sharp G band and 2D band can be seen in the as-synthesized graphite, higher graphitization degree of as-synthesized graphite compared with graphitic carbon synthesized by reacting CO_2 with CaC_2 at 700–800 °C.

Figure Left: Time dependence of temperature in the reactor during the reaction of CO_2 with 0.2 g LiAlH_4 . Right: Raman spectrum of as-synthesized graphite.

First-principles calculation was employed to further study the mechanism for the CO_2 pressure-dependent kinetics and thermodynamics of the reaction for synthesizing graphite. The energy difference between graphite and amorphous carbon is calculated to be -0.57 eV, supporting the favorable formation of graphite from the thermodynamic theory. A slight increase in total energy with a large increase in pressure is observed for both graphite and amorphous carbon (Supplementary Fig. 17). The total energy of the reactants of CO_2 and LiAlH_4 remarkably increases with the pressure, particularly for CO_2 (Supplementary Fig. 17). The density of CO_2 is dependent of its pressure due to the isometric process applied to synthesize graphite. For the CO_2 - LiAlH_4 system under low CO_2 pressure, an appreciable reduction in the

total energy of CO₂-LiAlH₄ system can be obtained by a small amount of CO₂ adsorbing on the LiAlH₄ surface (Supplementary Fig. 18). In order to achieve high pressure in the synthesis of graphite, gaseous CO₂ was compressed into a liquid, where the increased energy of CO₂ resulted from work done on gas approximately equals to the liquefaction heat of CO₂ of 0.16 eV [31]. The liquid CO₂ is converted into gaseous state during heating, in which the energy of CO₂ absorbed is equivalent to the vaporization heat of 0.16 eV [31]. The great increase in energy of 0.32 eV is the additional energy for the graphite synthesized under low CO₂ pressure. Overall, the CO₂ pressure-induced energy changes of reactants and products are responsible for the CO₂ pressure-dependent kinetics and thermodynamics of the reaction for synthesizing graphite.

The following content was added and updated:

Page 14, Para 1, Line 11,

...band and 2D band to D band increase to as high as 3.9 and 1.8, respectively, for the graphite submicroflakes. **It can be concluded that the kinetic barriers of the reaction for synthesizing graphite decrease with the gaseous back pressure.** In addition, CO₂ concentration has little effect on the **kinetic barriers of the reaction** for synthesizing graphite (Supplementary Fig. 9) in contrast with gaseous back pressure....

Page 14, Para 2,

Temperature is another key factor in synthesizing graphite from CO₂. Take the LiAlH₄ reacting with 35 bar CO₂ as an example, various reaction temperatures are achieved by adjusting the amounts of LiAlH₄ in the synthetic experiments. As the amounts of LiAlH₄ are increased from 0.3 g to 0.38 and 0.5 g, the maximum temperature increases from 471 °C to 752 and 960 °C, respectively (Supplementary Fig. 10). The 2D band is clearly seen in the Raman spectrum of the carbon synthesized at the maximum temperature of 960 °C (Supplementary Fig. 11) whereas it is not observed in the Raman spectra of the carbon synthesized at the maximum temperatures of 471 and 752 °C (Supplementary

Fig. 11). Moreover, their intensity ratios of G band to D band are 0.88, 1.01 and 1.47, respectively, signifying graphitization degree of carbon increases with temperature. The same conclusion can be drawn from the XRD patterns (Supplementary Fig. 11). The above Raman and XRD results indicate that the graphitization degree of carbons strongly depends on their temperatures. However, the kinetic barrier of a reaction is independent of temperature. To overcome the kinetic barriers, heating is a good way for reactants system to absorb energy. According to our first-principles molecular dynamics calculations, the total energy of CO₂ significantly increases with the temperature (Supplementary Fig. 12). This is the reason for the temperature-dependent synthetic reaction of graphite.

Figure S10 Time dependence of temperature in the reactor during 35 bar CO₂ reacting with LiAlH₄. (a) 0.30 g LiAlH₄, (b) 0.50 g LiAlH₄.

Figure S11 Raman spectra (a) and XRD patterns (b) of the carbon synthesized by

reacting 35 bar CO₂ with 0.3, 0.38 and 0.5 g LiAlH₄.

Figure S12 Effect of temperatures on the energy of CO₂ determined by first-principles molecular dynamics (FPMD) calculations.

Page 15, Para 2,

In order to analyze the reason for a small amount of amorphous carbon formed in the synthesis of graphite, carbon was synthesized by reacting LiAlH₄ with 2 bar CO₂ (Supplementary Fig. 13). The starting and maximum reaction temperature are 142 °C and 165 °C, respectively. The starting temperature is higher than the one in the high gaseous back pressures, but the maximum temperature is much lower than that in the high gaseous back pressures (Fig. 1b, Fig. 3d and Supplementary Fig. 8). The characteristic peaks of graphite are not observed in the XRD pattern of as-synthesized carbon (Supplementary Fig. 14). The G band almost overlaps the D band, and the 2D band is not seen in the Raman spectrum of the carbon synthesized under 2 bar CO₂ (Supplementary Fig. 15). The XRD and Raman results indicate that amorphous carbon is synthesized **but graphite is not synthesized under 2 bar CO₂**, resulted from the low CO₂ pressure-induced high kinetic barriers of the reaction for synthesizing graphite. The thermodynamic and kinetic competition of the reaction for

synthesizing graphite and amorphous carbon as illustrated in Supplementary Fig. 16 leads to the synthesis of graphite from CO₂ **strongly depended on CO₂ pressure and temperature. The kinetic barriers of the reaction for synthesizing graphite drop with rising of gaseous pressure. On the other hand, the kinetic barriers of the reaction for producing amorphous carbon is insensitive to gaseous pressure because the starting reaction temperature only decreased from 142 °C under 2.2 bar CO₂ to 126 °C under 132 bar CO₂ (Fig. 1b and Supplementary Fig. 13). Even under the high pressure of ~160 bar, the kinetic barriers of the reaction for synthesizing graphite are slightly higher than those for producing amorphous carbon while it is thermodynamically favorable to produce graphite.**

First-principles calculation based on density function theory was performed to study the CO₂ pressure-dependent kinetics and thermodynamics of the reaction for synthesizing graphite. The energy difference between graphite and amorphous carbon is calculated to be -0.57 eV, supporting the favorable formation of graphite from the thermodynamic theory. A slight increase in total energy with a large increase in pressure is observed for both graphite and amorphous carbon (Supplementary Fig. 17). The total energy of the reactants of CO₂ and LiAlH₄ remarkably increases with the pressure, particularly for CO₂ (Supplementary Fig. 17). The density of CO₂ is dependent of its pressure due to the isometric process applied to synthesize graphite. For the CO₂-LiAlH₄ system under low CO₂ pressure, an appreciable reduction in the total energy of CO₂-LiAlH₄ system can be obtained by a small amount of CO₂ adsorbing on the LiAlH₄ surface (Supplementary Fig. 18). In order to achieve high pressure in the synthesis of graphite, gaseous CO₂ was compressed into a liquid, where the increased energy of CO₂ resulted from work done on gas approximately equals to the liquefaction heat of CO₂ of 0.16 eV [31]. The liquid CO₂ is converted into gaseous state during heating, in which the energy of CO₂ absorbed is equivalent to the vaporization heat of 0.16 eV [31]. The great increase in energy of 0.32 eV is the additional energy for the graphite synthesized under low CO₂ pressure. Overall, the CO₂ pressure-induced energy changes of reactants and products are

responsible for the CO₂ pressure-dependent kinetics and thermodynamics of the reaction for synthesizing graphite.

Figure S17 Energy difference of graphite, amorphous carbon, LiAlH₄ and CO₂ with pressure.

Figure S18 Adsorption energies of sequentially adsorbed CO₂ molecules on the (100) surface LiAlH₄.

References:

- [31] John A. D. Section 6 Thermodynamic properties. *Lange's Handbook of Chemistry* (McGraw-Hill, Inc., 15th edition, 1998)

Responses to Reviewer 2#

Comment

This paper reports graphite submicroflakes prepared from CO₂ via reacting with LiAlH₄ at 126 °C. It is looking very nice at a glance. However, there are some serious issues which are needed to be clarified properly. This work can be reconsidered after a mandatory revision.

Reply:

Thank you for your positive comment. We have revised our manuscript according to your comments and the detailed responses are given in the **Comments 1 to 9**.

Comment 1

Reduction of CO₂ will produce graphite. Theoretically it is correct. But the author is silent about yield. The author should mention about the amount of precursor liquid CO₂ and how much CO₂ (%) is converting to graphite (yield).

Reply:

In our synthesis of graphite, CO₂ performs two functions. One is the reactant for synthesizing graphite. The other is to provide high gaseous pressure in the synthesis process of graphite as the temperature increases from room temperature to reaction temperatures. Therefore, an excess of CO₂ is applied in the synthesis process of graphite. In addition, hydrogen is produced in the synthesis process of graphite. It is difficult to determine the yield of graphite based on CO₂. On the contrary, the yield of graphite based on LiAlH₄ can be determined from the solid-state products of CO₂ reacting with LiAlH₄.

The yield of graphite based on LiAlH₄ has been added in the revised manuscript. According to TG measurement of the solid products of CO₂ reacting with LiAlH₄

heated under air, the yield of graphite based on LiAlH_4 is determined to be 20.7 wt.%, which is close to the theoretical value of 22.1 wt.% calculated from the chemical reaction (1). The yield of graphite is as high as 93.7%.

The following content was added:

Page 11, Para 1, the end,

... further demonstrate that the chemical interaction between CO_2 and LiAlH_4 can be expressed by Equation (1). **According to chemical reaction (1), the theoretical yield of graphite based on LiAlH_4 is calculated to be 22.1 wt.%, which corresponds closely to the experimental value of 20.7 wt.% determined by the TG measurement of solid products of CO_2 reacting with LiAlH_4 (Supplementary Fig. 5). The yield of graphite is as high as 93.7%.**

Comment 2

Even though the reaction temperatures are very low, gaseous pressure is too high. The advantage of mild synthesis condition is simple reactor design and low cost, but I could not find any reactor setup to sustain such a high pressure. The authors should clarify this issue related to high pressure.

Reply:

We all agree that the advantage of mild synthesis condition is generally simple reactor design and low cost for synthesizing a material. In this work, the reaction temperatures for synthesizing graphite are very low, the maximum gaseous pressure reaches 164 bar in the synthesis process of graphite with high degree of crystallization. For the laboratory reactor, the maximum pressure in the commercial reactors such as Parr 4590 or Büchi Limbo 450, may reach up to 345 bar. For the industrial reactors, the reactors can be designed on the basis of the hydrogen storage tanks. The maximum pressure in hydrogen storage tanks are as high as 700 bar. But the cost of high-pressure reactors may be not low.

In this work, the CO_2 pressure-dependent kinetic barriers of the reaction for synthesizing graphite was found for the first time. In order to decrease the gas pressure for synthesizing graphite, we may focus on the thermodynamic and kinetic

tuning of synthesis reactions via catalytic modification and composition design/optimization. Much effort will be needed to decrease the high pressure and the cost of reactors.

Comment 3

The proposed mechanism in session 2.3 is not clear. For instance, the formation of ordered meso phase as intermediate was observed? More strong evidences such as computational and in-situ analyses are required to confirm the mechanism, which should be further clarified more detailed.

Reply:

Thank you for your comments. The computational and experimental analyses are added in the revised manuscript. The in-situ experiments can't be performed because of the limited reaction time of 3 seconds. The first-principles calculation was employed to understand the formation mechanism of graphite.

The solid products of CO₂ reacting with LiAlH₄ (Fig. 1b) were heated at 880 °C for 3 h under argon. Few differences in graphitization degree is observed in the as-synthesized carbon with and without heat treatment at 880 °C (Supplementary Fig. 6), signifying that the amorphous carbon was not converted into graphite at 880 °C in the presence of LiAlO₂, Li₂CO₃ and Al. For understanding the graphitization of amorphous carbon, first-principles calculation was employed to calculate the kinetic barriers (Supplementary Fig. 7). The activation energy is as high as 1.66 eV, supporting that amorphous carbon was not converted into graphite at 880 °C and higher temperatures are needed for the graphitization of amorphous carbon.

In the original version, it has been concluded that the kinetic barriers of the reaction for synthesizing graphite decrease with the gaseous back pressure. Temperature is another key factor in synthesizing graphite from CO₂. The role of temperature in synthesizing graphite from CO₂ was discussed in the revised version. Take the LiAlH₄ reacting with 35 bar CO₂ as an example, various reaction temperatures are achieved by adjusting the amounts of LiAlH₄ in the synthetic experiments. As the amounts of LiAlH₄ are increased from 0.3 g to 0.38 and 0.5 g, the

maximum temperature increases from 471 °C to 752 and 960 °C, respectively (Supplementary Fig. 10). The 2D band is clearly seen in the Raman spectrum of the carbon synthesized at the maximum temperature of 960 °C (Supplementary Fig. 11) whereas it is not observed in the Raman spectra of the carbon synthesized at the maximum temperatures of 471 and 752 °C (Supplementary Fig. 11). Moreover, their intensity ratios of G band to D band are 0.88, 1.01 and 1.47, respectively, signifying graphitization degree of carbon increases with temperature. The same conclusion can be drawn from the XRD patterns (Supplementary Fig. 11). The above Raman and XRD results indicate that the graphitization degree of carbons strongly depends on their temperatures. However, the kinetic barrier of a reaction is independent of temperature. To overcome the kinetic barriers, heating is a good way for reactants system to absorb energy. According to our first-principles molecular dynamics calculations, the total energy of CO₂ significantly increases with the temperature (Supplementary Fig. 12). This is the reason for the temperature-dependent synthetic reaction of graphite.

The thermodynamic and kinetic competition of the reaction for synthesizing graphite and amorphous carbon as illustrated in Supplementary Fig. 16 leads to the synthesis of graphite from CO₂ strongly depended on CO₂ pressure and temperature. The kinetic barriers of the reaction for synthesizing graphite drop with rising of gaseous pressure. On the other hand, the kinetic barriers of the reaction for producing amorphous carbon is insensitive to gaseous pressure because the starting reaction temperature only decreased from 142 °C under 2.2 bar CO₂ to 126 °C under 132 bar CO₂ (Fig. 1b and Supplementary Fig. 13). Even under the high pressure of ~160 bar, the kinetic barriers of the reaction for synthesizing graphite are slightly higher than those for producing amorphous carbon while it is thermodynamically favorable to produce graphite.

First-principles calculation based on density function theory was performed to study the CO₂ pressure-dependent kinetics and thermodynamics of the reaction for synthesizing graphite. The energy difference between graphite and amorphous carbon is calculated to be -0.57 eV, supporting the favorable formation of graphite from the thermodynamic theory. A slight increase in total energy with a large increase in pressure is observed for both graphite and amorphous carbon (Supplementary Fig. 17). The total energy of the reactants of CO₂ and LiAlH₄ remarkably increases with the pressure, particularly for CO₂ (Supplementary Fig. 17). The density of CO₂ is dependent of its pressure due to the isometric process applied to synthesize graphite. For the CO₂-LiAlH₄ system under low CO₂ pressure, an appreciable reduction in the total energy of CO₂-LiAlH₄ system can be obtained by a small amount of CO₂ adsorbing on the LiAlH₄ surface (Supplementary Fig. 18). In order to achieve high pressure in the synthesis of graphite, gaseous CO₂ was compressed into a liquid, where the increased energy of CO₂ resulted from work done on gas approximately equals to the liquefaction heat of CO₂ of 0.16 eV [31]. The liquid CO₂ is converted into gaseous state during heating, in which the energy of CO₂ absorbed is equivalent to the vaporization heat of 0.16 eV [31]. The great increase in energy of 0.32 eV is the additional energy for the graphite synthesized under low CO₂ pressure. Overall, the CO₂ pressure-induced energy changes of reactants and products are responsible for the CO₂ pressure-dependent kinetics and thermodynamics of the reaction for synthesizing graphite.

The following content was added and updated:

Page 14, Para 1, Line 11,

...band and 2D band to D band increase to as high as 3.9 and 1.8, respectively, for the graphite submicroflakes. **It can be concluded that the kinetic barriers of the reaction for synthesizing graphite decrease with the gaseous back pressure.** In addition, CO₂ concentration has little effect on the **kinetic barriers of the reaction** for synthesizing graphite (Supplementary Fig. 9) in contrast with gaseous back pressure....

Temperature is another key factor in synthesizing graphite from CO₂. Take the LiAlH₄ reacting with 35 bar CO₂ as an example, various reaction temperatures are achieved by adjusting the amounts of LiAlH₄ in the synthetic experiments. As the amounts of LiAlH₄ are increased from 0.3 g to 0.38 and 0.5 g, the maximum temperature increases from 471 °C to 752 and 960 °C, respectively (Supplementary Fig. 10). The 2D band is clearly seen in the Raman spectrum of the carbon synthesized at the maximum temperature of 960 °C (Supplementary Fig. 11) whereas it is not observed in the Raman spectra of the carbon synthesized at the maximum temperatures of 471 and 752 °C (Supplementary Fig. 11). Moreover, their intensity ratios of G band to D band are 0.88, 1.01 and 1.47, respectively, signifying graphitization degree of carbon increases with temperature. The same conclusion can be drawn from the XRD patterns (Supplementary Fig. 11). The above Raman and XRD results indicate that the graphitization degree of carbons strongly depends on their temperatures. However, the kinetic barrier of a reaction is independent of temperature. To overcome the kinetic barriers, heating is a good way for reactants system to absorb energy. According to our first-principles molecular dynamics calculations, the total energy of CO₂ significantly increases with the temperature (Supplementary Fig. 12). This is the reason for the temperature-dependent synthetic reaction of graphite.

Figure S10 Time dependence of temperature in the reactor during 35 bar CO₂

reacting with LiAlH₄. (a) 0.30 g LiAlH₄, (b) 0.50 g LiAlH₄.

Figure S11 Raman spectra (a) and XRD patterns (b) of the carbon synthesized by reacting 35 bar CO₂ with 0.3, 0.38 and 0.5 g LiAlH₄.

Figure S12 Effect of temperatures on the energy of CO₂ determined by first-principles molecular dynamics (FPMD) calculations.

Page 15, Para 2,

In order to analyze the reason for a small amount of amorphous carbon formed in the synthesis of graphite, carbon was synthesized by reacting LiAlH₄ with 2 bar CO₂ (Supplementary Fig. 13). The starting and maximum reaction temperature are

142 °C and 165 °C, respectively. The starting temperature is higher than the one in the high gaseous back pressures, but the maximum temperature is much lower than that in the high gaseous back pressures (Fig. 1b, Fig. 3d and Supplementary Fig. 8). The characteristic peaks of graphite are not observed in the XRD pattern of as-synthesized carbon (Supplementary Fig. 14). The G band almost overlaps the D band, and the 2D band is not seen in the Raman spectrum of the carbon synthesized under 2 bar CO₂ (Supplementary Fig. 15). The XRD and Raman results indicate that amorphous carbon is synthesized **but graphite is not synthesized under 2 bar CO₂**, resulted from the low CO₂ pressure-induced high kinetic barriers of the reaction for synthesizing graphite. The thermodynamic and kinetic competition of the reaction for synthesizing graphite and amorphous carbon as illustrated in Supplementary Fig. 16 leads to the synthesis of graphite from CO₂ **strongly depended on CO₂ pressure and temperature. The kinetic barriers of the reaction for synthesizing graphite drop with rising of gaseous pressure. On the other hand, the kinetic barriers of the reaction for producing amorphous carbon is insensitive to gaseous pressure because the starting reaction temperature only decreased from 142 °C under 2.2 bar CO₂ to 126 °C under 132 bar CO₂ (Fig. 1b and Supplementary Fig. 13). Even under the high pressure of ~160 bar, the kinetic barriers of the reaction for synthesizing graphite are slightly higher than those for producing amorphous carbon while it is thermodynamically favorable to produce graphite.**

First-principles calculation based on density function theory was performed to study the CO₂ pressure-dependent kinetics and thermodynamics of the reaction for synthesizing graphite. The energy difference between graphite and amorphous carbon is calculated to be -0.57 eV, supporting the favorable formation of graphite from the thermodynamic theory. A slight increase in total energy with a large increase in pressure is observed for both graphite and amorphous carbon (Supplementary Fig. 17). The total energy of the reactants of CO₂ and LiAlH₄ remarkably increases with the pressure, particularly for CO₂ (Supplementary Fig. 17). The density of CO₂ is dependent of its pressure due to the isometric process applied to synthesize graphite. For the CO₂-LiAlH₄ system

under low CO₂ pressure, an appreciable reduction in the total energy of CO₂-LiAlH₄ system can be obtained by a small amount of CO₂ adsorbing on the LiAlH₄ surface (Supplementary Fig. 18). In order to achieve high pressure in the synthesis of graphite, gaseous CO₂ was compressed into a liquid, where the increased energy of CO₂ resulted from work done on gas approximately equals to the liquefaction heat of CO₂ of 0.16 eV [31]. The liquid CO₂ is converted into gaseous state during heating, in which the energy of CO₂ absorbed is equivalent to the vaporization heat of 0.16 eV [31]. The great increase in energy of 0.32 eV is the additional energy for the graphite synthesized under low CO₂ pressure. Overall, the CO₂ pressure-induced energy changes of reactants and products are responsible for the CO₂ pressure-dependent kinetics and thermodynamics of the reaction for synthesizing graphite.

Figure S17 Energy difference of graphite, amorphous carbon, LiAlH₄ and CO₂ with pressure.

Figure S18 Adsorption energies of sequentially adsorbed CO₂ molecules on the (100) surface LiAlH₄.

References:

[31] John A. D. Section 6 Thermodynamic properties. *Lange's Handbook of Chemistry* (McGraw-Hill, Inc., 15th edition, 1998)

Comment 4

In abstract it is claimed that “CO₂ is converted into graphite submicroflakes in the seconds timescale via reacting with LiAlH₄ as the CO₂–LiAlH₄ mixture is heated to 126 °C”... In conclusion it is claimed that “The graphite submicroflakes are successfully synthesized by reacting CO₂ with LiAlH₄ below 876 °C within”... Which one is correct temperature 126 or 876 °C? It is recommended to clearly mention about the claims as the synthesis procedure is the main finding of this work.

Reply:

Thank you for your suggestion. This description has been corrected in the revised manuscript.

The following content was updated:

Page 12, Para 3, Line 11,

... for the graphite being directly formed from CO₂ **at the temperature range of 126–876 °C** within several seconds (Fig. 1b).

Page 20, Line 2 from bottom,

... submicroflakes are successfully synthesized by reacting CO₂ with LiAlH₄ **at the temperature range of 126–876 °C** within several seconds in the absence ...

Comment 5

There are some other reports (Studies in Surface Science and Catalysis, 1998, 114, 147-152, J. Mater. Chem. A, 2015, 3, 21211-21218) of converting CO₂ to graphite. The author is suggested to compare these synthesis methods and to highlight the novelty of this work.

Reply:

According to reviewer's suggestion, our synthesis method has been compared with other reports on synthesis of graphite derived from CO₂ in the revised manuscript (J. Mater. Chem. A, 2015, 3, 21211-21218; Carbon, 2017, 116, 246-254). CO₂ can be converted into graphitic carbon sheets by molten salt electrolysis ((J. Mater. Chem. A, 2015, 3, 21211-21218) or thermal reaction of CO₂ with CaC₂ (Carbon, 2017, 116, 246-254). However, the graphitization degree of graphitic carbons is much less than our as-synthesized graphite since the intensity ratio of G band to D band of 3.9 is far greater than that of 1.7 of graphitic carbon synthesized by molten salt electrolysis at 850 °C and that of 1.3 of graphitic carbon synthesized by reacting CO₂ with CaC₂ at 700–800 °C. During the synthesis of above graphitic carbons, the evolution of CO is accompanied. In our work, the synthesis of graphite is a green, energy-conserving and time-saving process.

For the paper published in “Studies in Surface Science and Catalysis” (1998, 114, 147-152), CO₂ was reported to be reduced to graphitic carbon via methane by catalytic with membrane reactor. However, the characterization of graphitization degree of carbon was not found in the paper. As we known, the decomposition of methane is a industrial route to synthesize carbon black and hydrogen. Thus, this paper was not added in the revised manuscript.

The following content was added:

Page 8, Para 1, Line 5,

... CO₂ and hazard gases during the carbonization of precursors. **CO₂ has been reported to synthesize graphitic carbon sheets by molten salt electrolysis^[27] or thermal reaction of CO₂ with CaC₂^[28]. However, the graphitization degree of graphitic carbons is much less than our as-synthesized graphite since the intensity ratio of G band to D band of 3.9 is far greater than that of 1.7 of graphitic carbon synthesized by molten salt electrolysis at 850 °C and that of 1.3 of graphitic carbon synthesized by reacting CO₂ with CaC₂ at 700–800 °C. During the synthesis of above graphitic carbons, the evolution of CO is accompanied. In this work, the synthesis of graphite is a green, energy-conserving and time-saving process. Furthermore, our as-synthesized graphite is easy to separate from impurities ...**

References,

[27] Hu, L. W., Song, Y., Ge, J. B., Zhu, J., & Jiao S. Q. Capture and electrochemical conversion of CO₂ to ultrathin graphite sheets in CaCl₂-based melts. *J. Mater. Chem. A*, **3**, 21211–21218 (2015).

[28] He, R., Wang, Z. Y., & Jin X. B. Preparation of graphitic carbon nanosheets by reaction between CO₂ and CaC₂ for lithium-ion batteries. *Carbon* **116**, 246-254 (2017).

Comment 6

It is mentioned in introduction “the transition metal catalysts are found to be difficult to separate from synthetic graphite”... for other methods. However, it is not clear how pure graphite is obtained by the present technique, as the author also used LiAlH₄, KOH, hydrochloric acid etc.

Reply:

The purity of as-synthesized graphite was determined by ICP-MS, EDS and TG measurements in the revised manuscript. As shown in Figure S2, the solid residue was not detected for the as-synthesized graphite reacted with air. The C and O elements

are detected in the EDS and XPS measurements (Figure S1 and Figure 1). According to EDS result, the content of O is only 2.73 wt.% (Figure S1). For the chemical reagents and commercial graphite, the O element is usually not included in the purity of graphite. Therefore, the purity of as-synthesized graphite was characterized by ICP-MS (Table S1). The contents of Li, Al, K and other metal in as-synthesized graphite is determined to be 0.0038, 0.0034, 0.0030, and 0.0018 wt.%, corresponding to 99.988 wt.% (metals basis) of the purity of as-synthesized graphite. This purity is very close to 99.996 wt.% of the commercial graphite.

The following content was added:

Page 8, Para 1, the end,

... from impurities or byproducts as indicated by EDS analysis and thermogravimetric (TG) measurement (Supplementary Fig. 1 and Supplementary Fig. 2). **The content of metal ions in as-synthesized graphite was characterized by inductively coupled plasma–mass spectrometry (Supplementary Table 1). The purity of as-synthesized graphite is 99.988 wt.% (metals basis), very close to 99.996 wt.% of commercial graphite (Supplementary Table 1).**

Supporting information,

Table S1 Content of metal ions in as-synthesized graphite submicroflakes and commercial graphite

	Li/wt.%	Al/wt.%	K/wt.%	Other metals/wt.%
Graphite Submicroflakes	0.0038	0.0034	0.0030	0.0018
Commercial Graphite	0	0.00007	0.0026	0.0009

The content of metal ions was determined by ICP-MS.

Comment 7

How has the “carbon with low degree of graphitization” been removed with mixing with KOH?

Reply:

KOH as the activating reagent has been widely used in the chemical activation of various carbon sources. The KOH activation is a well-known method to generate the pore network in carbons because KOH can react with carbon in the activation procedures. After activation, porous carbons (amorphous carbon) show ultrahigh specific surface area up to $3000 \text{ m}^2 \text{ g}^{-1}$ [A. N. Wennerberg and T. M. O’Grady, U.S. Pat., 4,082,694, 1978]. Moreover, Cheng et al reported that the small pores on the surfaces of graphite disappear and large pores formed on the surface of graphite after KOH etching at $800 \text{ }^\circ\text{C}$ [Journal of Power Sources 284 (2015) 258-263]. The specific surface area of etched graphite is less than original graphite. These results support that the reaction of amorphous carbon with KOH is more easily than that of graphite with KOH. An amorphous carbon and as-synthesized graphite submicroflakes were treated with KOH at $850 \text{ }^\circ\text{C}$. The Raman spectra of amorphous carbon and graphite submicroflakes without and with KOH treatments are shown in the following figures. It can be clearly seen that both amorphous carbon and graphite submicroflakes exhibit higher degree of graphitization after KOH treatment, indicating that carbon with low degree of graphitization can be removed by KOH treatment.

Figure Upper: Raman spectra of amorphous carbon without and with KOH treatment; Lower: Raman spectra of as-synthesized graphite submicroflakes without and with KOH treatment.

Comment 8

The synthesis technique is quite confusing. In the main text, it mentioned that the graphite has been prepared in different pressure (35 bar and 75 bar) and temperatures. However, in the synthesis procedure it is not clearly reported about pressure and temperature.

Reply:

Thank you for your suggestion. The experimental section has been updated in the revised manuscript.

The following content was added and updated:

Supporting information, section 1.1 Materials and synthesis,

High-purity deionized (DI) water was prepared in our lab. The other chemical reagents were obtained from the commercial purchase and used as received. The graphite submicroflakes are synthesized by the chemical interaction between CO₂ and LiAlH₄ in a closed reactor. **For each synthetic experiment, 0.30, 0.38 or 0.50 g of**

lithium aluminium hydride (LiAlH₄, 97%) purchased from Alfa Aesar were loaded into a homemade reactor (30 mL) in a MBRAUN glovebox (O₂ < 0.5 ppm, H₂O < 0.5 ppm) filled with argon (99.995%, Outesen). **Then, gaseous CO₂ (99.995%, Pujiang) and/or argon was introduced into the above reactor. Various preset gaseous pressures were applied in the experiments. To produce a high pressure above 100 bar in the heating process, liquid CO₂ was into the above reactor via plunger pump because gaseous CO₂ was converted into the liquid above its critical pressure of 73.8 bar.** The CO₂–LiAlH₄ mixture ...

Supporting information, section 1.2 Characterization,

The products of CO₂ reacting with LiAlH₄ and graphite samples were characterized by X-ray diffraction (XRD, X'Pert PRO), Raman spectroscopy (Renishaw Invia plus), field emission scanning electron microscopy (FESEM, NOVA NANOSEM 450) **with energy dispersive spectroscopy (EDS, Oxford X-Max 80 SDD)**, high-resolution transmission electron microscopy (HRTEM, FEI Tecnai G2 F30), **inductively coupled plasma–mass spectrometry (ICP-MS, PerkinElmer Elan DRC-e)**, and **X-ray photoelectronic spectroscopy (XPS, Kratos Axis Ultra DLD) with monochromatized Al K α excitation source.** The XRD data ...

...to determine the specific surface area and pore size distribution of graphite submicroflakes. **Thermogravimetric analysis (TG, Q5000IR) was carried out from room temperature to 700 °C a heating rate of 5 °C min⁻¹ in air.** Gas composition was ...

Comment 9

It is recommended to further analyze the chemical structure of the as-synthesized material by XPS.

Reply:

The as-synthesized graphite submicroflakes were characterized by XPS in the revised manuscript. XPS analysis shows that the intensity of sp²-C peak is much greater than that of sp³-C peak as shown in the high-resolution XPS spectrum of C 1s.

This result indicates the as-synthesized graphite submicroflakes have been further demonstrated to be high degree of graphitization.

The following content was added:

Page 6, Para 3,

...is observed in the Raman spectrum of graphite. **The as-synthesized graphite can be further demonstrated to be high degree of graphitization by X-ray photoelectron spectra (Fig. 1e and f). A very small amount of O was detected in the as-synthesized graphite, in which O is chemical bonding with C. For the C element, the intensity of sp^2 -C peak is much greater than that of sp^3 -C peak as shown in the high-resolution XPS spectrum of C 1s.** Figure 2 presents the SEM, TEM, HRTEM and SAED images of as-synthesized graphite....

Supporting information, Section 1.2 Characterization

The products of CO_2 reacting with $LiAlH_4$ and graphite samples were characterized by X-ray diffraction (XRD, X'Pert PRO), Raman spectroscopy (Renishaw Invia plus), field emission scanning electron microscopy (FESEM, NOVA NANOSEM 450) **with energy dispersive spectroscopy (EDS, Oxford X-Max 80 SDD)**, high-resolution transmission electron microscopy (HRTEM, FEI Tecnai G2 F30), **inductively coupled plasma-mass spectrometry (ICP-MS, PerkinElmer Elan DRC-e)**, and **X-ray photoelectronic spectroscopy (XPS, Kratos Axis Ultra DLD) with monochromatized Al $K\alpha$ excitation source.** The XRD data were collected on...

The Figure 1 was updated as follows:

Figure 1 The synthesis and characterization of graphite derived from CO₂. (a) Schematic illustration of the synthesis of graphite submicroflakes. (b) Time dependence of temperature in the reactor during the reaction process. (c) XRD pattern of the solid products after reaction. (d) Raman spectrum of the solid products after the removal of impurities. (e) XPS survey spectrum of graphite submicroflakes. (f) High-resolution XPS spectrum of C 1s.

Reviewer #1 (Remarks to the Author):

The authors have addressed all concerns of this referee. It became clearly that the pressure could be an more essential factor for the synthesis of graphite, but this important information has not been reflected in the abstract. Therefore, a minor suggestion of this referee is to have the pressure condition of synthesis in the abstract.

Reviewer #2 (Remarks to the Author):

This revised manuscript has resolved most of concerns that I claimed. However, some points are still unclear. For instance, there are no details of reactor setup to show how to sustain such a high pressure. Although the term of "mild" was removed, "energy-saving" is still emphasized. I don't agree with that this reaction is energy saving system because high pressure requires energy intensive system and operating conditions. And information about the mechanistic study by computational analysis is not provided in detail. For instance, Figure S16 is schematically illustrated without any calculated value and detailed pathway. Therefore, this work can not be accepted yet unless all concerns are fully resolved.

Responses to Reviewer 1#

Comment

The authors have addressed all concerns of this referee. It became clearly that the pressure could be a more essential factor for the synthesis of graphite, but this important information has not been reflected in the abstract. Therefore, a minor suggestion of this referee is to have the pressure condition of synthesis in the abstract.

Reply:

Thank you for your suggestion. We have revised the abstract of our manuscript according to your comments.

The following content was added:

Abstract:

...the lowest temperature with the shortest time for synthesizing graphite up to now. **Gas pressure-dependent kinetic barriers for synthesizing graphite is demonstrated to be the major reason for our synthesis of graphite without the graphitization process of amorphous carbon.** When serving as lithium storage materials...

Responses to Reviewer 2#

Comment

This revised manuscript has resolved most of concerns that I claimed. However, some points are still unclear. For instance, there are no details of reactor setup to show how to sustain such a high pressure. Although the term of "mild" was removed, "energy-saving" is still emphasized. I don't agree with that this reaction is energy saving system because high pressure requires energy intensive system and operating conditions. And information about the mechanistic study by computational analysis is not provided in detail. For instance, Figure S16 is schematically illustrated without any calculated value and detailed pathway. Therefore, this work can not be accepted

yet unless all concerns are fully resolved.

Reply:

Thank you for your comments. We have added the details of our home-made reactor setup and how to sustain the high pressure. According to your comments, the computational analysis was added and the term of “energy-saving” is deleted in the revised manuscript.

Figure S0 Schematic diagram of home-made reactor setup and CO₂ injection system.

As shown in Figure S0, our home-made stainless-steel reactor setup is mainly composed of a sample cell with a sample temperature monitor and a gas reservoir with temperature and pressure monitors. The volume is 30 and 20 ml, respectively, for the sample cell and the gas reservoir in this work. The sample cell is connected to the gas reservoir by gas line. The temperatures and gas pressures can be recorded by an online data collection system. The sample in the sample cell can be heated by ovens, but the gas reservoir is maintained at room temperature during overall experimental process. The thickness of stainless-steel tubes for manufacturing reactors is dependent of the maximum pressure of the designed reactor setup.

After sample loaded into reactor, liquid CO₂ was then injected into reactor by plunger pump. The gaseous CO₂ pressure in compressed cylinders is higher than 50 bar at room temperature. The gaseous CO₂ will become liquid state at 31.2 °C when

the pressure is higher than the critical pressure of 73.8 bar. In this work, the plunger pump is used to compress gaseous CO₂ from the gas cylinders up to above its critical pressure, which is also applied in the supercritical CO₂ extraction technique. Valves 1 and 3 were opened in the CO₂ injection process and were closed as the CO₂ injection was finished. Valve 2 was kept close in the CO₂ injection process whereas it was kept open to connect the reactor to gas reservoir in the subsequent heating and reaction process of CO₂ with LiAlH₄. The important function of gas reservoir was to act as a pressure buffer to prevent reactors against overload in the subsequent heating and reaction process.

Figure 1b Time dependence of temperature and pressure in the reactor during the reaction process of CO₂ with LiAlH₄.

Gas and liquid CO₂ coexist in the reactor setup at room temperature. The pressure values as shown in Figure 1b are the pressures of gas-state CO₂. The CO₂ pressure in the reactor setup was gradually increased from ~75 bar at 35 °C to ~132 bar at 126 °C via furnace heating. The exothermic reaction of CO₂ with LiAlH₄ occurred at 126 °C and the furnace heating is stopped at the temperature above 126 °C. The steep increase in gas pressure from ~132 to ~168 bar is resulted from the sample temperatures increased from 126 to 876 °C in 3 seconds owing to the chemical heat release of the exothermic reaction of CO₂ with LiAlH₄. The reactor cooling was carried out in the furnace without electrical heating after the exothermic reaction of CO₂ with LiAlH₄. Therefore, the changes in gas pressure in the reactor setup can be summarizes as follows: (1) gas pressure at room temperature due to CO₂ injection, (2) gas pressure at below 126 °C increased by furnace heating, (3) gas pressure at the

temperature range of 126-876 °C increased by chemical heat release, (4) gas pressure decreased by natural cooling.

As for the term of “energy-saving”, it was deleted in the revised manuscript according to the reviewer’s comments.

Information about the mechanistic study by computational analysis has been provided in detail in the revised manuscript. For the Figure S16, it schematically illustrates the effect of gas pressure on the kinetic barrier of the chemical reaction of CO₂ with LiAlH₄ for synthesizing graphite and amorphous carbon. Figure S16 was plotted based on the experimental results combined with DFT calculation. The energy of products (graphite/amorphous carbon, LiAlO₂, Li₂CO₃, Al and H₂) is lower than that of reactants (CO₂-LiAlH₄ system) since the exothermic nature was demonstrated for the synthesis reaction of graphite/amorphous carbon. Among carbon allotropes, graphite is the most thermodynamically stable allotrope under standard conditions. Our DFT calculation shows that the energy difference between amorphous carbon and graphite is -0.57 eV/formula, implying that graphite is more thermodynamically stable than amorphous carbon. The DFT result is in good agreement with above conclusion. Transition-state theory, also called activated-complex theory, treatment of chemical reactions and other processes that regards them as proceeding by a continuous change in the relative positions and potential energies of the constituent atoms and molecules. In this work, the DFT calculation can’t treat the chemical reaction for synthesizing graphite from CO₂ because this synthesis reaction is the complicated reactions about bulk materials. The energy difference between transition state and reactants as shown in Figure S16, which is also named kinetic barrier, was estimated by the following experimental results. The initial CO₂ pressure increased from 2 bar to 35 bar, 75 bar (liquid CO₂) at room temperature. During heating, the corresponding CO₂ pressure at the initial reaction temperature was determined to be 2.2 bar at 142 °C (Figure S13), 42 bar at 128 °C (Figure 3d), and 132 bar at 126 °C (Figure 1b), respectively. The amorphous carbon was formed at the initial stage of synthesis reaction under low and high pressures. The above results indicate the kinetic barrier of the synthesis reaction of amorphous carbon is gradually reduced with the

CO₂ pressure. Moreover, the graphite with good crystalline was only produced at the second stage of synthesis reaction under high pressure, in which the maximum pressure is 168 bar at maximum temperature of 876 °C (Figure 1b), the kinetic barrier for synthesizing graphite can be remarkably reduced under high gas pressure. Compared with amorphous carbon, the reduction in kinetic barrier for synthesizing graphite is much greater than that of amorphous carbon whereas the kinetic barrier for synthesizing graphite is still higher than that for synthesizing amorphous carbon. The mechanism for CO₂ pressure-dependent kinetics and thermodynamics of the reaction for synthesizing graphite was studied by DFT calculation in the manuscript.

Besides, the computational details on the models of amorphous carbon, graphite and LiAlH₄ have been also added in the revised manuscripts.

In Manuscript:

The following content was updated and added:

Title:

The title “Green **and energy-saving** synthesis of graphite from CO₂ without graphitization process of amorphous carbon” was changed to “Green synthesis of graphite from CO₂ without graphitization process of amorphous carbon”.

Abstract:

The sentence “Herein, we report a green, **energy-saving** and efficient approach of synthesizing graphite from CO₂ at ultralow temperatures in the absence of transition metal catalysts.” was changed to “Herein, we report a green and efficient approach of synthesizing graphite from CO₂ at ultralow temperatures in the absence of transition metal catalysts.”.

Page 4, Para 1, Line 4:

The sentence “The green, **energy-saving** and efficient synthesis of graphite with controllable microstructure and morphology remains a considerable challenge.” was changed to “The green and efficient synthesis of graphite with controllable microstructure and morphology remains a considerable challenge.”.

Page 8, Para 1, Line 12:

The sentence “In this work, the synthesis of graphite is a green, **energy-conserving** and time-saving process.” was changed to “In this work, the synthesis of graphite is a green, and time-saving process.”.

Page 20, Line 4 from bottom:

The sentence “we have demonstrated a new method for the green **and energy-saving** synthesis of graphite from CO₂ at low temperatures.” was changed to “we have demonstrated a new method for the green synthesis of graphite from CO₂ at low temperatures.”.

In Supporting Information:

The following content was updated and added:

1.1 Materials and synthesis (SI Page 2)

High-purity deionized (DI) water was prepared in our lab. The other chemical reagents were obtained from the commercial purchase and used as received. The graphite submicroflakes are synthesized by the chemical interaction between CO₂ and LiAlH₄ in a **home-made stainless-steel reactor setup (Figure S0)**. **The reactor setup is mainly composed of a sample cell with a sample temperature monitor and a gas reservoir with temperature and pressure monitors. The volume is 30 and 20 ml, respectively, for the sample cell and the gas reservoir. The sample cell is connected to the gas reservoir by gas line. The temperatures and gas pressures can be recorded by an online data collection system.** For each synthetic experiment, 0.30, 0.38 or 0.50 g of lithium aluminium hydride (LiAlH₄, 97%) purchased from Alfa Aesar were loaded into a **sample cell** in a MBRAUN glovebox (O₂ < 0.5 ppm, H₂O < 0.5 ppm) filled with argon (99.995%, Outesen). Then, gaseous CO₂ (99.995%, Pujiang) and/or argon was introduced into the above reactor **setup**. Various preset gaseous pressures were applied in the experiments. To produce a high pressure above 100 bar in the heating process, liquid CO₂ was into the above reactor via a **gas injection system (Figure S0)** because gaseous CO₂ was converted into the liquid above its critical pressure of 73.8 bar. **Valves 1 and 3 were opened in the CO₂ injection process and were closed as the CO₂ injection was finished. Valve 2 was**

kept close in the CO₂ injection process whereas it was kept open to connect the reactor to gas reservoir in the subsequent heating and reaction process of CO₂ with LiAlH₄. The CO₂-LiAlH₄ mixture was heated at...

Figure S0 Schematic diagram of home-made reactor setup and CO₂ injection system.

1.4 Computational details (SI Page 5)

...For geometry optimization, the atomic coordinates were relaxed until the Hellmann-Feynman forces were less $0.01 \text{ eV}/\text{\AA}$. **The structural models of graphite (space group: P63/MMC), LiAlH₄ (space group: P2₁/C), CO₂ (space group: Pa-3) and amorphous carbon were employed to calculate the pressure dependent energy differences as shown in Fig. S17. The structural model of amorphous carbon was quenched and optimized from the melted carbon at 6000 K with a supercell of 64 carbon atoms by first-principles molecular dynamics (FPMD) calculation with the canonical ensemble (NVT). The Nose-Hoover thermostat was used to control the temperature. In order to calculate...**

...On the other hand, **FPMD calculations at 300 K, 400 K, 500 K and 700 K** were performed to study the temperature effect on the CO₂. At each temperature, MD simulation with a period of 10 ps **and a time step of 1 fs** was performed with the canonical ensemble NVT as well....

2. Figures (SI Pages 14 to 16)

The Figures S16 and 17 were updated and a description was added:

Figure S16 Schematic illustration of thermodynamics and kinetics of the reaction for synthesizing graphite and amorphous carbon from CO₂. **This schematic illustration was plotted based on the experimental results combined with DFT calculation.** The energy of products (graphite/amorphous carbon, LiAlO₂, Li₂CO₃, Al and H₂) is lower than that of reactants (CO₂-LiAlH₄ system) since the exothermic nature was demonstrated for the synthesis reaction of graphite/amorphous carbon. Our DFT calculation shows that the energy difference between amorphous carbon and graphite is -0.57 eV/formula, implying that graphite is more thermodynamically stable than amorphous carbon. The energy difference between transition state and reactants was estimated by the following experimental results. The initial CO₂ pressures are 2 bar to 35 bar, 75 bar (liquid

CO₂) at room temperature for synthesizing carbon. During heating, the corresponding CO₂ pressure at the initial reaction temperature was determined to be 2.2 bar at 142 °C (Figure S13), 42 bar at 128 °C (Figure 3d), and 132 bar at 126 °C (Figure 1b), respectively. The amorphous carbon was formed at the initial stage of synthesis reaction under low and high pressures. The above results indicate the kinetic barrier of the synthesis reaction of amorphous carbon is gradually reduced with the CO₂ pressure. Moreover, the graphite with good crystalline was only produced at the second stage of synthesis reaction under high pressure, in which the maximum pressure is 168 bar at maximum temperature of 876 °C (Figure 1b), the kinetic barrier for synthesizing graphite can be remarkably reduced under high gas pressure. Compared with amorphous carbon, the reduction in kinetic barrier for synthesizing graphite is much greater than that of amorphous carbon whereas the kinetic barrier for synthesizing graphite is still higher than that for synthesizing amorphous carbon.

Figure S17 The structural models of LiAlH₄, CO₂, Graphite and amorphous-C, and their pressure dependent energy differences. The green, grey, white, brown and red balls in the structural models represent the Li, Al, H, C and O atoms, respectively.